# Learning Task Belief Similarity with Latent Dynamics for Meta-Reinforcement Learning

**Menglong Zhang, Fuyuan Qian, Quanying Liu**[*]
Southern University of Science and Technology
{zhangml2022, qianfy2023}@mail.sustech.edu.cn
liuqy@sustech.edu.cn

## Abstract

Meta-reinforcement learning requires utilizing prior task distribution information obtained during exploration to rapidly adapt to unknown tasks. The efficiency of an agent's exploration hinges on accurately identifying the current task. Recent Bayes-Adaptive Deep RL approaches often rely on reconstructing the environment's reward signal, which is challenging in sparse reward settings, leading to suboptimal exploitation. Inspired by bisimulation metrics, which robustly extracts behavioral similarity in continuous MDPs, we propose SimBelief—a novel meta-RL framework via measuring similarity of task belief in Bayes-Adaptive MDP (BAMDP). SimBelief effectively extracts common features of similar task distributions, enabling efficient task identification and exploration in sparse reward environments. We introduce latent task belief metric to learn the common structure of similar tasks and incorporate it into the specific task belief. By learning the latent dynamics across task distributions, we connect shared latent task belief features with specific task features, facilitating rapid task identification and adaptation. Our method outperforms state-of-the-art baselines on sparse reward MuJoCo and panda-gym tasks.

## 1 Introduction

In meta-reinforcement learning, an agent is required to efficiently explore the environment to gather information relevant to the current task and use that information to adapt to new, unseen tasks (Duan et al., 2016; Finn et al., 2017; Gupta et al., 2018b; Humplik et al., 2019). However, in real-world scenarios, there are many distractions, and irrelevant information can cause the agent to engage in erroneous exploration behaviors, leading to the learning of a suboptimal policy. This challenge is exacerbated when the exploration space is vast, and rewards are sparse. Humans, on the other hand, have the ability to generalize across similar tasks by quickly identifying common patterns, such as recognizing that both opening a window and a drawer involve a pulling action. These shared structures can be abstracted and applied to new tasks through certain transformations, enabling rapid adaptation by leveraging prior knowledge. This process involves identifying similar features and relationships between tasks, with the underlying *task belief* facilitating effective knowledge transfer (Figure 1).

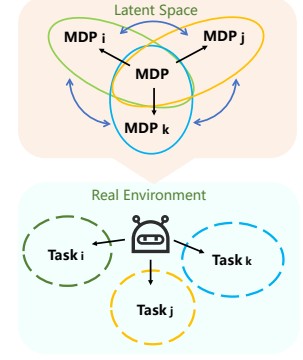

Figure 1: Learning shared structures among tasks in the latent space as task beliefs enables the agent to rapidly adapt to new tasks.

Common approaches to task belief modeling include posterior sampling (Thompson, 1933; Osband et al., 2013; Rakelly et al., 2019) and Bayes-Adaptive Markov Decision Processes (BAMDPs) (Duff, 2002; Ghavamzadeh et al., 2015), where BAMDPs provide a structured framework for learning and adapting to new tasks by integrating prior information (Zintgraf et al., 2019). Previous works have addressed the issue of

---

[*]Corresponding author.

uninformative rewards in meta-RL by using intrinsic rewards (Zhang et al., 2021c) to extract task-relevant information or exploration bonuses (Zintgraf et al., 2021) to enhance performance in hard exploration tasks. However, they often neglect the potential common structures shared across tasks.

To efficiently learn the structural relationships embedded within task belief and address the challenges of effective task identification and rapid adaptation, it is necessary to develop methods that can efficiently recognize tasks with similar structures. This requires exploring strategies beyond BAMDP-based approaches, which may involve leveraging hierarchical reinforcement learning (Frans et al., 2018), or task embeddings (Liu et al., 2021; Lan et al., 2019) that can capture the underlying structure of tasks and facilitate transfer learning across related tasks.

In this work, we propose **SimBelief** (Learning **Sim**ilarity of Task **Belief** for Meta-RL) as a meta-RL framework under BAMDP to measure the difference of tasks in similar distributions by learning the dynamics and transferring capability in latent space. This results in a **task belief similarity**, which quantifies the representational relationships between any two random tasks in the learned latent space. However, directly reconstructing and exploring the environment in the latent space may lose crucial detailed information (Yarats et al., 2021; Kemertas & Aumentado-Armstrong, 2021; Hafner et al., 2019), which is important for accurately reconstructing the specific task and ensuring convergence to the Bayesian optimal policy(Zintgraf et al., 2019; Choshen & Tamar, 2023).

Therefore, we combine the task belief similarity learned in the latent space, which provides an overall understanding of the task distribution and the relationships between tasks, with the belief of the specific task currently being learned, significantly enhancing the agent's adaptability to unknown environments. Compared to related meta-RL methods, our approach effectively learns dynamics in the latent space and leverages the learned latent representations to quickly discern relationships between tasks. This enables the agent to utilize prior knowledge from tasks, recognize the current environment, and accelerate exploration and adaptation to unseen tasks.

Our main contributions are as follows:

- We propose a task representation method for context-based meta-reinforcement learning algorithms in BAMDP, which enhances the agent's ability to recognize and adapt to unknown tasks by learning task belief similarity through latent reward model, transition model, and inverse dynamics model.

- In scenarios with sparse or uninformative reward signals, our algorithm more effectively extracts latent task space representations compared to other baselines and achieves state-of-the-art performance on sparse reward MuJoCo and panda-gym tasks.

- Our algorithm demonstrates stronger adaptability and generalization capabilities to out-of-distribution (OOD) tasks.

- We theoretically validate the effectiveness of the *latent task belief metric* in Bayes-Adaptive Markov Decision Processes (BAMDPs).

## 2 BACKGROUND

### 2.1 PROBLEM FORMULATION

In meta-reinforcement learning (meta-RL), we consider a distribution $p(\mathcal{T})$ over a space of tasks $\mathcal{T}$, where each task is modeled as a partially observable Markov decision process (POMDP; (Cassandra et al., 1994)) defined as $\mathcal{T} = (\mathcal{S}, \mathcal{A}, \mathcal{O}, P, R, \rho_0, \gamma, H)$. Here, $\mathcal{S}$ and $\mathcal{A}$ represent the shared state and action spaces across all tasks, while $\mathcal{O}$ denotes the task-specific observation space. The transition function $P(s' \mid s, a)$, reward function $R(r \mid s, a)$, and initial state distribution $\rho_0$ define the task-specific dynamics.

For context-based methods, the agent interacts with the environment over a horizon of $H$ time steps in each episode of each task and utilizes a task-specific context $\tau_t = (o_{1:t}, a_{1:t-1}, r_{1:t-1})$ to infer the hidden task dynamics. The objective is to learn a policy $\pi(a_t \mid \tau_t)$ that can rapidly adapt to new tasks by leveraging the learned task belief encoded in the context. The full horizon comprises $N$ task episodes, referred to as a *meta-episode*, each with its own set of $T$ time steps, and the task context is updated as the agent gathers more observations. The objective in meta-RL is to maximize

the expected return:

$$J(\pi) = \mathbb{E}_{\mathcal{T} \sim p(\mathcal{T})} \left[ \sum_{t=1}^{H} \mathbb{E}_{(o_t, a_t, r_t) \sim \pi} [r_t] \right]. \tag{1}$$

To address the multi-task adaptation problem in POMDPs, it is essential to avoid local optima and balance exploration with exploitation. Utilizing the framework of Bayes-Adaptive Markov Decision Processes (BAMDPs; (Duff, 2002)), we can represent uncertainty over task-relevant information in a belief space. By leveraging the current belief, we can reconstruct the dynamics model to better handle uncertainty, enabling more effective adaptation to the current task. The BAMDP is defined as $M^+ = (\mathcal{S}^+, \mathcal{A}, \rho_0^+, P^+, R^+, H^+)$, where the hyper-state space $\mathcal{S}^+ = \mathcal{S} \times \mathcal{B}$ combines the state space $\mathcal{S}$ with the task belief space $\mathcal{B}$, representing the posterior over MDPs based on past experiences. $P^+$ is the transition function, $\rho_0^+$ is the initial hyper-state distribution, and $R^+$ is the reward function. The total decision horizon is $H^+ = N \times H$. The goal is to learn a meta-policy $\pi^+(a_t \mid s_t^+)$ that maximizes cumulative reward by balancing exploration of task uncertainty and exploitation of the current belief.

## 2.2 BISIMULATION METRICS IN RL

Bisimulation metrics provide a way to measure state similarity in reinforcement learning (RL) based on behavioral equivalence, focusing on transitions and rewards rather than raw state features. The concept originated in model checking (Givan et al., 2003). In RL, bisimulation aggregates states that lead to similar future outcomes. This allows for more efficient learning. Approximate bisimulation metrics (Ferns et al., 2004; Castro, 2020) extended this idea to high-dimensional environments, allowing RL agents to better generalize, reduce state space complexity, and improve exploration. These metrics are particularly useful in tasks with sparse rewards or partial observability (Guo et al., 2022).

**Definition 1** ($\pi$-Bisimulation Metric (Castro, 2020)). *Given two states $s_i$ and $s_j$ from the state space S, the distance between these states can be evaluated using the $\pi$-bisimulation metric $d_\pi(s_i, s_j)$, which measures behavioral similarity. We define the metric $d_\pi(s_i, s_j)$ as:*

$$d_\pi(s_i, s_j) = |R_{s_i}^\pi - R_{s_j}^\pi| + \gamma W_1(d_\pi)(P_{s_i}^\pi, P_{s_j}^\pi) \tag{2}$$

*where $R_{s_i}^\pi$ and $R_{s_j}^\pi$ represent the reward functions under policy , $P^\pi$ denotes the transition distributions, and $W_1$ is the Wasserstein distance between the distributions.*

## 2.3 UNINFORMATIVE REWARDS IN META-REINFORCEMENT LEARNING

Uninformative rewards provide insufficient feedback for agents to quickly learn effective strategies (Andrychowicz et al., 2017; Dulac-Arnold et al., 2019). This issue can impede exploration, slow convergence, and lead to suboptimal policies. To address this challenge, methods such as intrinsic motivation and curiosity-driven exploration have been employed to encourage agents to explore state spaces even when extrinsic rewards are minimal (Pathak et al., 2017; Burda et al., 2018). Reward shaping techniques modify the reward function to offer more informative feedback, thereby accelerating learning (Ng et al., 1999). Balancing the exploration-exploitation trade-off is also crucial for efficient learning (Sutton, 2018). Enhancing meta-RL agents' ability to learn from uninformative rewards can significantly improve their performance across complex, real-world environments by incorporating advanced exploration strategies and reward augmentation methods (Gupta et al., 2018b).

In this work, we learn the dynamics of multiple tasks in a latent space and use a task belief metric to measure the differences between similar tasks. By extracting similar structures within the latent space and representing these structures as task belief similarity, we map them to the current real and unknown environment to assist the agent in exploration. This approach aims to achieve rapid adaptation to multiple tasks in environments with uninformative rewards.

## 3 METHOD

In this section, we provide a detailed introduction to SimBelief (Figure 2), an effective off-policy meta-RL approach for online adaptation within the BAMDP framework. First, we propose a latent

task belief metric to quantify the relationships between different tasks in the latent space in section 3.1. Then, we explain how task belief similarity is learned within the learned latent dynamics in section 3.2. We also describe the overall algorithmic process of SimBelief in detail in section 3.3. Finally, we theoretically prove the conditions under which the policy can transfer between tasks in the latent space in section 3.4.

## 3.1 LATENT TASK BELIEF METRIC

Previous work (Zhang et al., 2021b) learns an environment encoder to capture the relationships between Block MDPs (Du et al., 2019) for multi-task generalization. In (Zhang et al., 2021c), information gain is used as an intrinsic reward for exploration. However, in these approaches, the agent needs to know the current task information (i.e., task ID) during training. In contrast, in context-based meta-RL, these approaches limit adaptability to out-of-distribution tasks, as the agent must infer the task being executed from historical information and reason about the current task distribution during adaptation. Including task ID during training can partially constrain the agent's reasoning capabilities. In real-world scenarios, relying solely on the environment's inherent reward signal or manually designed intrinsic rewards often fails to effectively capture the common structure of tasks (Zheng et al., 2020). Therefore, learning the structural similarities between tasks and leveraging belief to facilitate rapid transfer across similar tasks is more advantageous in complex environments, especially those with sparse reward signals. To measure the relationships between tasks and make it applicable within the BAMDP, we define the *latent task belief metric*.

**Definition 2** (Latent Task Belief Metric). *Given two latent task beliefs $b_i$ and $b_j$, with corresponding samples $z_i$ and $z_j$ identified by their latent representations, let $d_\pi$ denote the metric that evaluates the distance between these two task beliefs. We define $d_\pi(z_i, z_j)$ as follows:*

$$d_\pi(z_i, z_j) = \left| R_i^\pi(s_i^+, a_i) - R_j^\pi(s_j^+, a_j) \right| + W_2(d_\pi) \left( T_i^\pi(s_i^+, a_i), T_j^\pi(s_j^+, a_j) \right)$$
$$+ \left\| I_i^\pi(s_i^+, s_i'^+) - I_j^\pi(s_j^+, s_j'^+) \right\|_1 \tag{3}$$

*where $s_i^+ = (g(s_i), z_i)$ is the augmented state corresponding to the task $i$, $r_i = R_i^\pi(s_i^+, a_i)$ is the reward model, $s_i'^+ = T_i^\pi(s_i^+, a_i)$[1] is the transition model, and $a_i = I_i^\pi(s_i^+, s_i'^+)$ is the inverse dynamics model.*

Here, we map $s$ to a latent compressed space as $g(s)$, and learn the task dynamics in the compressed space. Latent dynamic space can be defined as $\mathcal{S}^+ = \mathcal{G}(\mathcal{S}) \times \mathcal{Z}$. In Figure 2, $z_l$ represents the sample from latent task belief $b_l$ in the latent space and captures the similarity between different tasks.[2] All task distributions share a common latent dynamic space, with different tasks distinguished by different $z_l$, where $z_l$ contains the dynamic similarity information between any two tasks. We employ the inverse dynamics model to predict $a$, as it not only helps retain information relevant to what the agent can control but also enhances the agent's reasoning capability (see Appendix G.2). Unlike in Definition 1, here we use the dynamic model to measure task belief similarity to enhance exploration efficiency in unknown environments, while preserving the agent's control information in sparse reward settings. This approach is more effective for identifying similar tasks.

## 3.2 LEARNING TASK BELIEF SIMILARITY IN LATENT SPACE

Our goal is to use the latent task belief metric to capture the similarity information between any two tasks, which is embedded within the task belief. This enables SimBelief to achieve rapid task reasoning and adaptation. From the replay buffer $\mathcal{D}$, we randomly sample trajectories $\tau$ from two tasks (i.e., task $i$ and task $j$) in the task set $M$. The trajectory is then fed into the context encoder $q_\phi$ to obtain historical information $h^i$ and $h^j$, which serve as prior knowledge for the tasks and are input into the latent space. The latent space primarily consists of three components: (1) a state encoder $g_\theta$ that maps the original state space to the latent state space $\mathcal{S}^+$, (2) a belief similarity learner $\psi_l$ that maps $h$ to the latent task belief $b_l$, and (3) a latent dynamics model $p_\theta$ conditioned on the learned latent task belief (Figure 2).

---

[1] $s'$ represents the state at the next time step.

[2] For simplicity, in the latent task belief metric, $z_i$, $z_j$ and $z_l^i$, $z_l^j$ are used interchangeably to represent the same meaning.

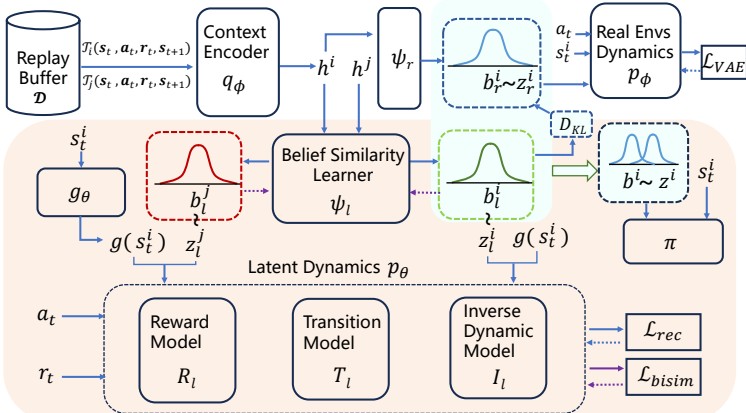

Figure 2: SimBelief architecture. Our framework consists of three components: learning latent task belief similarity, reconstructing the specific task, and the policy. In the compressed state space, $\psi_l$ learns the *latent task belief* $b_l^i$ through latent dynamics shared among all tasks in the distribution. In the real environment, we reconstruct the dynamics of the specific task using $\psi_r$ to obtain the *specific task belief* $b_r^i$. The policy $\pi$ is conditioned on the state and the combined belief $b^i$, which integrates $b_r^i$ and $b_l^i$. In this paper, we use $l$ and $r$ to denote the latent space and real space, respectively, and $i$ and $j$ to represent task $i$ and task $j$, respectively.

To obtain the latent dynamic representation of all tasks conditioned on $z_l$, we reconstruct the latent dynamics $p_\theta((s'^+, r, a) \mid z_l)$, which includes the reward model $R_l$, the transition model $T_l$, and the inverse dynamics model $I_l$ through one-step predictions. The learning objective for the latent dynamics can be expressed as:

$$\mathbb{E}_{p(M)}\left[\mathbb{E}_{\psi_l(z_l|h)}\left[\log p_\theta((s'^+, r, a) \mid z_l)\right]\right]. \tag{4}$$

This objective enables the model to capture task-specific latent dynamics by maximizing the likelihood of predicting the next augmented state $s'^+$, reward $r$, and action $a$ given the latent task belief $z_l$.

We aim to learn the similarity relationship between any two tasks using the latent task belief metric and the latent dynamics learned through Equation 4. This similarity relationship is obtained by optimizing the belief similarity learner $\psi_l$. The learning objective for task belief similarity using $\psi_l$ is defined as follows:

$$\mathcal{L}_{bisim}(\psi_l) = \|\psi_l(h_i) - \psi_l(h_j)\|_1 - \left|\hat{R}(s_i^+, a_i) - \hat{R}(s_j^+, a_j)\right| - W_2\left(\hat{T}(\cdot \mid s_i^+, a_i), \hat{T}(\cdot \mid s_j^+, a_j)\right)$$
$$- \left\|\hat{I}(\cdot \mid s_i^+, s_i'^+) - \hat{I}(\cdot \mid s_j^+, s_j'^+)\right\|_1, \tag{5}$$

where the gradients of dynamics model are stopped, and $b_l^i = \psi_l(h_i)$. We use the 2-Wasserstein metric because the $W_2$ metric has a convenient closed-form solution (Zhang et al., 2021a). During training, through Equation 4 and 5, $z_l^i$, sampled from $b_l^i$, is optimized to encode both the latent dynamic information required to reconstruct task $i$ and the similarity information between task $i$ and any other task in the task set $M$. Through this approach, we can utilize $\psi_l$ to identify contextual information at the current timestep and represent the similarity between any tasks through its output $b_l$.

### 3.3 SIMBELIEF ALGORITHM

We now proceed to describe how the learned task belief similarity (Section 3.2) can be leveraged to enhance the exploration capability and convergence efficiency of meta-RL. The algorithm pseudocode can be found in Appendix C.

**Reconstructing the specific task.** In the BAMDP framework (Ghavamzadeh et al., 2015; Zintgraf et al., 2019), even with extremely sparse rewards, it is necessary to construct the specific task's reward model in the original state space using a variational auto-encoder (VAE, (Kingma, 2013)) . This is because reconstructing the specific task in the latent state space may result in the loss of crucial information, as the latent space is solely used for learning task belief similarity. Additionally,

along the temporal dimension, we only reconstruct the past history, as we use a one-step prediction approach in the latent space. This approach effectively captures local information about the task distribution, enhancing the agent's reasoning ability for future predictions (see the upper part of Figure 2). The objective is to maximise

$$\mathcal{L}_{\text{VAE}}(\phi) = \mathbb{E}_{q_\phi(z_r|\tau_{:t})}\left[\log p_\phi(\tau_{:t} \mid z_r)\right] - D_{\text{KL}}\left(q_\phi(z_r \mid \tau_{:t}) \parallel p(z_r)\right), \tag{6}$$

where $q_\phi$ generates the specific task belief $z_r$ for the current timestep using only the information from the historical trajectory $\tau_{:t}$. We consider $q_\phi$ as the forward reasoning process to infer the task belief $z_r$, corresponding to $z_r \sim \psi_r(b_r \mid h)q_\phi(h \mid \tau_{:t})$ in the Algorithm 1.

**Integrating latent task belief for effective online exploration.** The learned latent task belief $z_l \sim b_l$ enables the agent to quickly recall the similarities between different tasks during interactions with the environment, enhancing the efficiency of reasoning and exploration. This is the primary reason why SimBelief achieves robust adaptation to OOD tasks (Figure 4). Since the latent state space is an abstraction of the original task, the latent task belief may overlook the specific task information required by the agent during exploration in the real environment. Therefore, we choose to project the two distributions into a joint space $\mathcal{B}$, and then use a Gaussian mixture of the two distributions $b_r = (\mu_{\psi_r}(h), \sigma_{\psi_r}(h))$ and $b_l = (\mu_{\psi_l}(h), \sigma_{\psi_l}(h))$, combined with the real environment's augmented state derived from $s_t$, as the input to the policy $\pi$. The formulation of the Gaussian mixture distribution $b$ is given by:

$$b = w_r \cdot \mathcal{N}(z_r \mid \mu_r, \sigma_r^2) + w_l \cdot \mathcal{N}(z_l \mid \mu_l, \sigma_l^2), \tag{7}$$

where $\mathcal{N}(z \mid \mu_r, \sigma_r^2)$ and $\mathcal{N}(z \mid \mu_l, \sigma_l^2)$ are Gaussian distributions with means $\mu_r$ and $\mu_l$, and variances $\sigma_r^2$ and $\sigma_l^2$, respectively. The weights $w_r$ and $w_l$ satisfy $w_r + w_l = 1$ and are non-negative (see Appendix G.3).

In the early stages of training, the latent task belief enhances the agent's ability to identify tasks and explore efficiently in the real environment, providing a high-level understanding of the overall task distribution (see Appendix F.2). However, for the algorithm to converge stably, it need to incorporate finer-grained information about specific tasks, which is captured in the specific task belief $b_r$. Therefore, we minimize the discrepancy between the latent belief $b_l$ and the specific task belief $b_r$ when training $\psi_l$. This is achieved by introducing a KL divergence term between the two beliefs, serving as a regularization for the latent task belief. Combined with Equation 5, the overall optimization objective of $\psi_l$ can be written as:

$$\mathcal{J}(\psi_l) = \mathbb{E}_{p(M)}\left[\mathcal{L}_{bisim}(\psi_l) + D_{\text{KL}}\left(\hat{q}_\phi(z_r \mid \tau_{:t}) \parallel p(z_l)\right)\right], \tag{8}$$

where the gradient of $\hat{q}_\phi$ is stopped, and $p(z_l)$ represents the distribution of $z_l \sim \psi_l(b_l \mid h)$.

SimBelief utilizes Soft Actor-Critic (SAC) (Haarnoja et al., 2018) to optimize the policy, where the optimization of latent dynamics and latent task similarity is conducted jointly with policy optimization to ensure convergence to near Bayes-optimal policy. To enhance the consistency between latent dynamics and real environment exploration, without introducing perturbations that could disrupt belief similarity in the latent space, we use the loss of Q-function as the objective to train the offset $(\Delta\mu, \Delta\sigma)$ of $b_l = (\mu_{\psi_l}(h) + \Delta\mu, \sigma_{\psi_l}(h) + \Delta\sigma)$ (see Appendix G.4). The loss of SAC after integrating latent dynamic information is

$$\mathcal{J}(\pi) = \mathbb{E}_{(s,a)\sim\mathcal{D}}\left[\alpha \log\left(\pi(a \mid (s,b))\right) - Q_\theta((s,b),a)\right], \tag{9}$$

where $(s, b)$ is the augmented state in the real environment under the BAMDP framework.

### 3.4 THEORETICAL ANALYSIS

Based on the latent task belief metric $d_\pi(z_i, z_j)$ defined in Definition 2, we can establish the following theorems regarding the difference in task dynamics between two tasks in the latent space within BAMDP.

**Theorem 1** (Value difference bound). *Given two tasks $\mathcal{M}_i$ and $\mathcal{M}_j$ in the latent space with states $s_i^+, s_j^+ \in S^+$, and let $V^\pi$ be the value function of policy $\pi$, the value difference bound between the tasks can be given by:*

$$|V^\pi(s_i^+) - V^\pi(s_j^+)| \leq d_\pi(z_i, z_j), \tag{10}$$

*where $d_\pi(z_i, z_j)$ is the latent task belief metric.*

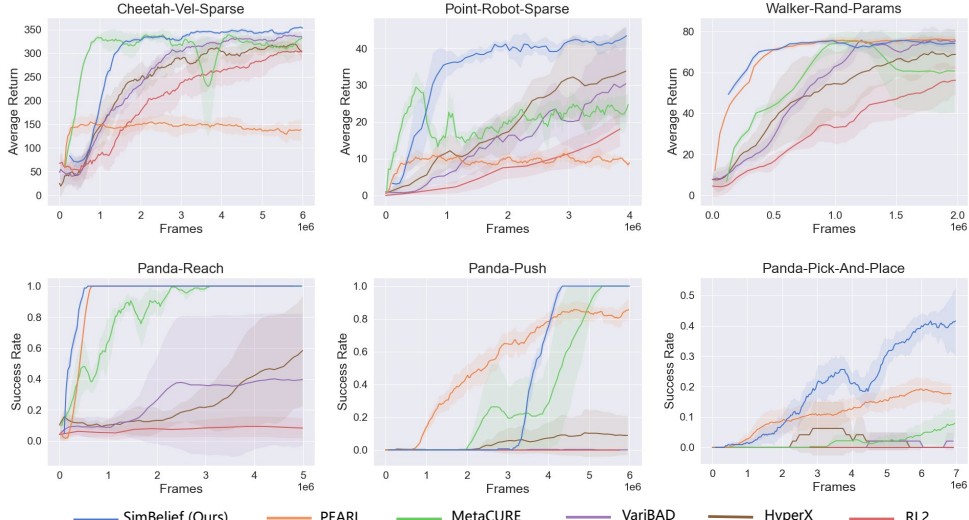

Figure 3: Meta-testing performance on sparse reward MuJoCo and panda-gym tasks over 3 random seeds. Our algorithm, SimBelief, demonstrates superior online adaptation capabilities compared to other algorithms.

This theorem demonstrates that tasks with similar latent task beliefs $z_i$ and $z_j$ will have similar value function under the same policy $\pi$.

**Theorem 2** (Latent transfer bound). *Let $Q^*_{\mathcal{M}_j}$ be the optimal Q-function for task $\mathcal{M}_j$. The difference between $Q^*_{\mathcal{M}_j}$ and the Q-function of the policy $\pi$ learned from task $\mathcal{M}_i$, applied to task $\mathcal{M}_j$, is bounded as follows:*

$$\left\| Q^*_{\mathcal{M}_j} - [Q^\pi_{\mathcal{M}_i}]_{\mathcal{M}_j} \right\|_\infty \le \epsilon_R + \gamma \left( \epsilon_T + \epsilon_I + \|z_i - z_j\|_1 \right) \frac{R_{max}}{2(1-\gamma)}. \tag{11}$$

This extended transfer bound theorem provides a formal bound on how well a policy $\pi$ trained on task $\mathcal{M}_i$ will transfer to task $\mathcal{M}_j$. $\epsilon_R$, $\epsilon_T$, and $\epsilon_I$ are approximation errors for rewards, transitions, and inverse dynamics. All proofs are provided in Appendix B.

## 4 EXPERIMENTS

In this section, we evaluate SimBelief on sparse-reward tasks in MuJoCo (Finn et al., 2017; Rakelly et al., 2019) and the more challenging panda-gym (Gallouédec et al., 2021) environment. We aim to address the following questions: 1. Can SimBelief achieve fast online adaptation in sparse reward tasks? 2. Can SimBelief leverage learned latent belief similarity representations to enhance out-of-distribution generalization? 3. What is the impact of latent task representations on rapid exploration? 4. How does the latent space correspond to the real environment?

**Environments and baselines:** We conducted experiments on six complex sparse reward tasks, including Point-Robot-Sparse, Cheetah-Vel-Sparse, Walker-Rand-Params, Panda-Reach, Panda-Push, and Panda-Pick-And-Place (see Appendix E). While MuJoCo tasks are commonly used by current meta-learning algorithms, panda-gym simulates real-world robotic arm movements with extremely sparse rewards, making it a better benchmark for evaluating an algorithm's ability to handle real-world tasks. We compared SimBelief against PEARL (Rakelly et al., 2019), which is based on posterior sampling, and MetaCURE (Zhang et al., 2021c), which learns a separate policy, as well as VariBAD (Zintgraf et al., 2019), HyperX (Zintgraf et al., 2021), and RL[2] (Duan et al., 2016), which are based on the BAMDP framework (see Appendix D).

**Online Adaptation Performance.** During the training phase, we performed meta-testing by calculating the meta-episode average return and success rate across different tasks to evaluate the algorithm's online performance. As shown in Figure 3, SimBelief consistently performed well across all tasks and exhibited superior adaptation capabilities compared to other algorithms. BAMDP-based algorithms often require more interactions with the environment to converge to a Bayes-optimal

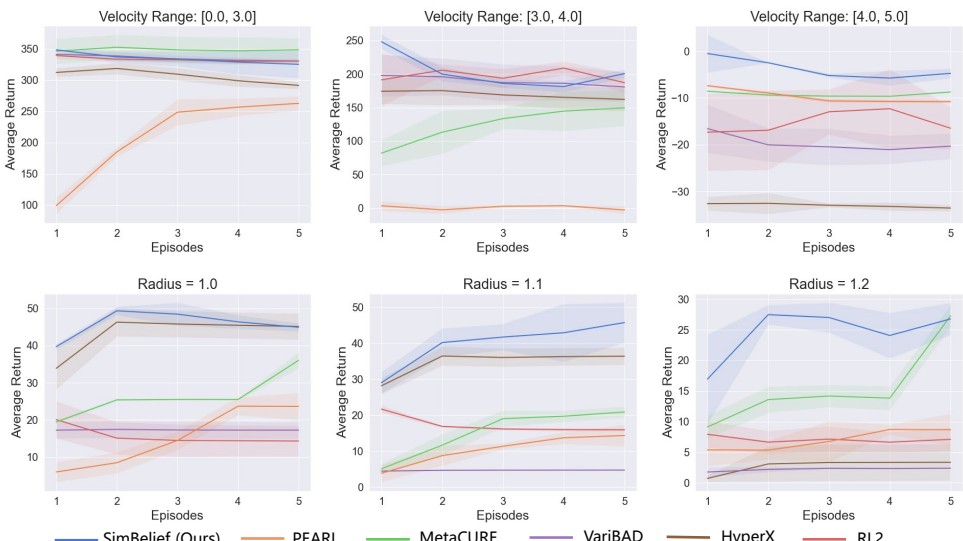

Figure 4: Average test performance for the first 5 rollouts on **out-of-distribution** tasks (Cheetah-Vel-Sparse and Point-Robot-Sparse) shows that as the task distribution gradually deviates from the training distribution, SimBelief demonstrates more efficient and robust adaptation compared to other algorithms, while still maintaining a high return.

policy, whereas our algorithm, by integrating task belief similarity in the latent space, significantly improved the agent's ability to effectively extract task-relevant information in sparse reward environments, thus accelerating convergence. In sparse reward tasks like those in panda-gym, the performance of Varibad, HyperX, and RL$^2$ illustrates that overly relying on the environment's reward signal while neglecting task similarity information makes it difficult to achieve online adaptation in more complex scenarios. Additionally, SimBelief demonstrated more stable performance compared to posterior sampling-based algorithms like PEARL and MetaCURE, indicating that SimBelief accurately learned the common structure of the task distribution.

**Out-of-distribution Task Inference and Adaptation.** We evaluated the algorithm's reasoning and generalization capabilities on out-of-distribution (OOD) tasks. In Figure 4, we show the agent's out-of-domain adaptation and generalization performance on the Cheetah-Vel-Sparse (top) and Point-Robot-Sparse (bottom) tasks. The leftmost column represents the task range used during the training phase, where the agent was trained by generating task goals within this range. During the testing phase, we adjusted the velocity range for Cheetah-Vel-Sparse and the semi-circle radius for Point-Robot-Sparse to assess the agent's robustness and task inference capabilities within five episodes. The results indicate that SimBelief exhibits stable out-of-distribution generalization, demonstrating that learned latent belief similarity representations can be leveraged to enhance out-of-distribution task inference ability. The latent task belief similarity we proposed also equips the agent with strong task transferability.

**Exploration Performance.** We visualized the exploration and adaptation process of the agent in the Point-Robot-Sparse environment over two episodes, as shown in Figure 5. The test target appeared within a semicircle with a radius of 1.2, and five positions on the semicircle were randomly selected as the agent's targets. Compared to other algorithms, SimBelief successfully identified the randomly appearing target location within the semicircle and controlled the robot to reach the target point within two episodes. Moreover, in the out-of-distribution setting, SimBelief demonstrated more efficient learning of a near Bayes-optimal policy compared to BAMDP-based algorithms like HyperX. The latent task belief enabled the agent to efficiently analogize and identify tasks, leading to more effective exploration.

**Task Belief Visualization.** We use t-SNE (Van der Maaten & Hinton, 2008) to visualize SimBelief's latent task belief and specific task belief (Figure 6), aim to illustrate how the task common structure learned in the latent space contributes to effective reconstruction of the real environment. This visualization reveals the underlying reason for SimBelief's ability to achieve rapid out-of-distribution generalization. The latent space, as an abstraction and holistic representation of the task distri-

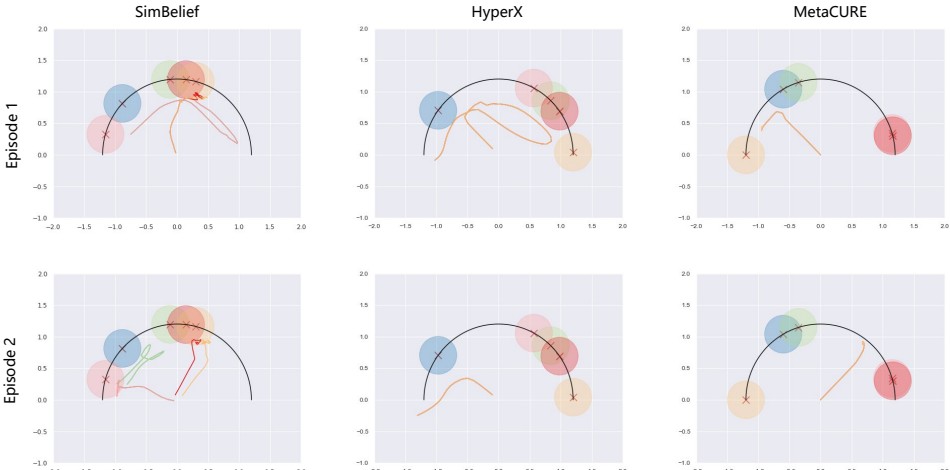

Figure 5: Exploration and adaptation performance in the 5 random **out-of-distribution (radius = 1.2) tasks of Point-Robot-Sparse**. SimBelief is capable of robustly learning a near Bayes-optimal policy.

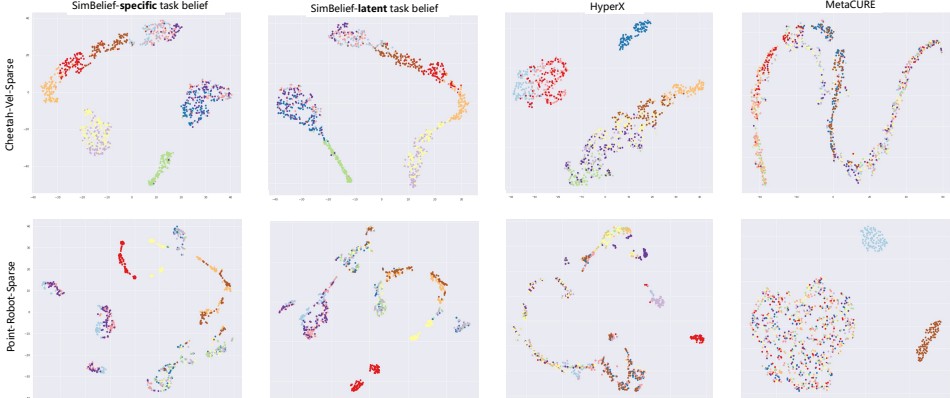

Figure 6: t-SNE visualization of task beliefs learned by the algorithm on 10 randomly sampled OOD tasks. The velocity range for Cheetah-Vel-Sparse is set to 3-4, and the semicircle radius for Point-Robot-Sparse is 1.1. SimBelief demonstrates a more distinct representation in the latent space, enhancing the specific task belief to represent similar tasks in the real environment.

bution, enables efficient transfer of essential task information. This, in turn, allows the agent to leverage prior knowledge for identifying and reasoning quickly about unknown environments.

**The mechanism of SimBelief for rapid adaptation.** SimBelief, through the latent task belief, essentially learns the transfer relationships between knowledge across different tasks. In extremely sparse reward scenarios, once the agent succeeds in one task, this prior knowledge of success can be quickly propagated to other similar tasks via the latent task belief, enabling the agent to rapidly adapt to similar types of tasks. The latent task belief acts as a bridge, facilitating the generalization of past successes to new, related tasks, significantly reducing exploration time and computational resources. A more detailed discussion is provided in Appendix F.

## 5   RELATED WORKS

**Task representation in reinforcement learning** focuses on how agents encode and utilize task information to enhance learning efficiency and generalization. Meta-reinforcement learning enables agents to quickly adapt to new tasks by learning task priors; for instance, (Finn et al., 2017) introduced Model-Agnostic Meta-Learning (MAML) for rapid adaptation with minimal updates, while

(Rakelly et al., 2019) proposed PEARL to learn latent task representations for fast probabilistic adaptation. (Zintgraf et al., 2019; Dorfman et al., 2021) learn a latent variable model of task distribution for efficient Bayes-adaptive RL, and (Gupta et al., 2018a) explored unsupervised meta-learning without explicit task labels. Yuan & Lu (2022); Choshen & Tamar (2023) apply contrastive learning to enhance encoder representation. (Lee et al., 2023) decomposes complex tasks into subtasks to handle non-parametric task variability. In multi-task RL, (Teh et al., 2017) introduced Distral to learn shared policies across tasks with task-specific adaptations. Contextual task representations have also been explored: (Sodhani et al., 2021) use metadata to learn interpretable representations, and (Hausman et al., 2018) learned embedding spaces for transferable skills. These studies underscore the importance of efficient task representation to improve RL performance and generalization.

**Exploration with task inference.** Integrating exploration with task inference in reinforcement learning enables agents to learn efficient policies by understanding task structures. Information-theoretic approaches encourage exploration by maximizing the entropy of state visitation distributions, promoting diverse behaviors (Liu & Abbeel, 2021). (Raileanu & Rocktäschel, 2020) propose a type of intrinsic reward which encourages the agent to take actions that lead to significant changes in its learned state representation. (Wan et al., 2023) introduce DEIR, a method that enhances exploration efficiency and robustness by using discriminative model-based episodic intrinsic rewards. (Yang et al., 2023) propose an unsupervised skill discovery method through contrastive learning among behaviors. (Rana et al., 2023) propose a low-level residual policy for skill adaptation enabling downstream RL agents to adapt to unseen tasks Planning-based methods enable agents to plan exploratory actions that maximize information gain by learning world models (Sekar et al., 2020). (Xie et al., 2020) leverage the idea of partial amortization for fast adaptation at test time.

**Bisimulation for control.** Bisimulation has been effectively applied in various control tasks, demonstrating its value in reducing computational burdens without sacrificing policy performance (Zhang et al., 2021a). (Zhang et al., 2021b) propose a method for learning state abstractions that generalize across tasks, improving sample efficiency and performance in multi-task and meta-reinforcement learning by leveraging shared dynamics in complex environments. (Gelada et al., 2019) has leveraged bisimulation for efficient exploration by ensuring the agent treats bisimilar states similarly, reducing sample complexity. (Hansen-Estruch et al., 2022) propose a form of state abstraction that captures functional equivariance for goal-conditioned RL. (Kemertas & Aumentado-Armstrong, 2021) generalize value function approximation bounds for on-policy bisimulation metrics to non-optimal policies. These studies highlight the increasing applicability of bisimulation in reinforcement learning and control tasks. They offer a variety of solutions that reduce sample complexity, guide exploration, and enhance learning performance, particularly in goal-conditioned settings and beyond.

# 6 CONCLUSION

We present SimBelief, a meta-RL framework aimed at improving the agent's ability to quickly adapt and generalize in real-world sparse reward tasks. At the core of SimBelief is the latent task belief metric, which learns the similarity between the dynamics of various tasks in a shared latent space and represents this similarity as task belief similarity. This helps the agent efficiently capture the common structure of the task distribution, facilitating the recognition and reasoning about unknown tasks. By combining the latent task belief with the specific task belief learned through interactions in the real environment, SimBelief demonstrates strong adaptation and generalization capabilities during exploration, particularly on out-of-distribution tasks. It overcomes the slow convergence and inefficiency issues inherent in BAMDP. We believe that SimBelief will inspire future research on agent generalization, providing a powerful tool for enabling agents to more efficiently tackle the challenges and complexities of real-world environments.

ACKNOWLEDGMENTS

This work wasfunded in part by National Natural Science Foundation of China (62472206), Shenzhen Excellent Youth Project (RCYX20231211090405003), Shenzhen Science and Technology Innovation Committee (2022410129, KJZD20230923115221044, KCXFZ20201221173400001).

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

# A    THEORETICAL BACKGROUND AND ANALYSIS

In this section, we introduce the theoretical background and the rationale behind the design of the latent task belief metric.

**Definition 3** (Bisimulation Relation (Givan et al., 2003)). *Given an MDP $\mathcal{M}$, an equivalence relation $E \subseteq S \times S$ is a bisimulation relation if whenever $(s, t) \in E$, the following properties hold, where $S_E$ is the state space $S$ partitioned into equivalence classes defined by $E$:*

$$\forall a \in A, \mathcal{R}(s, a) = \mathcal{R}(t, a) \tag{12}$$

$$\forall a \in A, \forall c \in S_E, \mathcal{P}(s, a)(c) = \mathcal{P}(t, a)(c), where \mathcal{P}(s, a)(c) = \sum_{s' \in c} \mathcal{P}(s, a)(s') \tag{13}$$

**Definition 4** (Diffuse Metric). *A diffuse metric measures the distance between two points by considering not only the shortest path but also the distribution of multiple paths between the points. The diffuse metric satisfies the following properties:*

*1. Non-negativity: For any two points $x$ and $y$, $d(x, y) \geq 0$, and $d(x, y) = 0$ if and only if $x = y$.*
*2. Symmetry: For any two points $x$ and $y$, $d(x, y) = d(y, x)$.*
*3. Triangle inequality: For any three points $x$, $y$, and $z$, $d(x, z) \leq d(x, y) + d(y, z)$.*

*These properties ensure that the diffuse metric behaves as a valid distance measure across the space.*

To prove that the metric $d_\pi(z_i, z_j)$ in Defination 2 is valid, we show it satisfies three key properties: non-negativity, symmetry, and the triangle inequality.

*Proof.* Each term in the metric $d_\pi(z_i, z_j)$ is non-negative: $|R_i^\pi(s_i^+, a_i) - R_j^\pi(s_j^+, a_j)|$ is the absolute difference in rewards, which is non-negative. $W_2(T_i^\pi(s_i^+, a_i), T_j^\pi(s_j^+, a_j))$ is the Wasserstein distance, which is non-negative. $\|I_i^\pi(s_i^+, s_{i+1}^+) - I_j^\pi(s_j^+, s_{j+1}^+)\|_1$ is an $L_1$-norm, which is non-negative. Thus, $d_\pi(z_i, z_j) \geq 0$. Additionally, all three types of distances satisfy symmetry, hence $d_\pi(z_i, z_j) = d_\pi(z_j, z_i)$.

To derive the triangle inequality for latent task belief metric, we need to verify that for any three task beliefs $z_i$, $z_j$, and $z_k$, the following holds:

$$d_\pi(z_i, z_k) \leq d_\pi(z_i, z_j) + d_\pi(z_j, z_k)$$

Expand the distance $d_\pi(z_i, z_j)$ and $d_\pi(z_j, z_k)$

$$d_\pi(z_i, z_j) = |R_i^\pi(s_i^+, a_i) - R_j^\pi(s_j^+, a_j)| + W_2(d_\pi)(T_i^\pi(s_i^+, a_i), T_j^\pi(s_j^+, a_j)) + \|I_i^\pi(s_i^+, s_i'^+) - I_j^\pi(s_j^+, s_j'^+)\|_1$$

and similarly,

$$d_\pi(z_j, z_k) = |R_j^\pi(s_j^+, a_j) - R_k^\pi(s_k^+, a_k)| + W_2(d_\pi)(T_j^\pi(s_j^+, a_j), T_k^\pi(s_k^+, a_k)) + \|I_j^\pi(s_j^+, s_j'^+) - I_k^\pi(s_k^+, s_k'^+)\|_1$$

Expand the distance $d_\pi(z_i, z_k)$

$$d_\pi(z_i, z_k) = |R_i^\pi(s_i^+, a_i) - R_k^\pi(s_k^+, a_k)| + W_2(d_\pi)(T_i^\pi(s_i^+, a_i), T_k^\pi(s_k^+, a_k)) + \|I_i^\pi(s_i^+, s_i'^+) - I_k^\pi(s_k^+, s_k'^+)\|_1$$

To prove the triangle inequality, we need to show that:

$$|R_i^\pi(s_i^+, a_i) - R_k^\pi(s_k^+, a_k)| \leq |R_i^\pi(s_i^+, a_i) - R_j^\pi(s_j^+, a_j)| + |R_j^\pi(s_j^+, a_j) - R_k^\pi(s_k^+, a_k)|$$

This follows directly from the standard triangle inequality for absolute values. Similarly, for the Wasserstein distance term:

$$W_2(d_\pi)(T_i^\pi(s_i^+, a_i), T_k^\pi(s_k^+, a_k)) \leq W_2(d_\pi)(T_i^\pi(s_i^+, a_i), T_j^\pi(s_j^+, a_j)) + W_2(d_\pi)(T_j^\pi(s_j^+, a_j), T_k^\pi(s_k^+, a_k))$$

This holds due to the triangle inequality for the Wasserstein distance (Villani et al., 2009). Finally, for the inverse dynamics term:

$$\|I_i^\pi(s_i^+, s_i'^+) - I_k^\pi(s_k^+, s_k'^+)\|_1 \leq \|I_i^\pi(s_i^+, s_i'^+) - I_j^\pi(s_j^+, s_j'^+)\|_1 + \|I_j^\pi(s_j^+, s_j'^+) - I_k^\pi(s_k^+, s_k'^+)\|_1$$

This also follows directly from the standard triangle inequality for norms. Since each of the three components satisfies the triangle inequality, we can conclude that the latent task belief metric $d_\pi$ also satisfies the triangle inequality:

$$d_\pi(z_i, z_k) \leq d_\pi(z_i, z_j) + d_\pi(z_j, z_k)$$

□

## B  THEOREMS AND PROOFS

**Lemma 1.** *Let $\mathcal{S}^+ = \mathcal{G}(\mathcal{S}) \times \mathcal{Z}$ be the latent dynamics space, and $\pi$ is the improving policy. The $\mathcal{F}$ function is defined as follows:*

$$
\begin{aligned}
\mathcal{F}(d,\pi)(z_i, z_j) = & |R_i^\pi(s_i^+, a_i) - R_j^\pi(s_j^+, a_j)| + W_2(d_\pi)(T_i^\pi(s_i^+, a_i), T_j^\pi(s_j^+, a_j)) \\
& + \|I_i^\pi(s_i^+, s_{i+1}^+) - I_j^\pi(s_j^+, s_{j+1}^+)\|_1,
\end{aligned} \tag{14}
$$

*where $s_i^+ = (g(s_i), z_i)$, $s_j^+ = (g(s_j), z_j)$ are the augmented states corresponding to task $i$ and task $j$, and $z_i$, $z_j$ represent different task beliefs in latent space $\mathcal{S}^+$. Then, there exists a unique least fixed point $\tilde{d}$ such that $\mathcal{F}(\tilde{d}) = \tilde{d}$.*

*Proof.* To prove that $\mathcal{F}$ has a unique least fixed point, we need to establish that $\mathcal{F}$ is monotonic and continuous, ensuring the conditions for a least fixed-point result.

We first show that $\mathcal{F}$ is monotonic. Suppose $d \leq d'$, meaning $d(z_i, z_j) \leq d'(z_i, z_j)$ for all $z_i, z_j$ in latent space. Then, by the definition of $\mathcal{F}$, we have:

$$
W_2(d_\pi)(T_i^\pi(s_i^+, a_i), T_j^\pi(s_j^+, a_j)) \leq W_2(d_\pi')(T_i^\pi(s_i^+, a_i), T_j^\pi(s_j^+, a_j)).
$$

Since the Wasserstein distance is non-decreasing in $d$, it follows that $\mathcal{F}(d, \pi) \leq \mathcal{F}(d', \pi)$. Additionally, the reward difference and inverse dynamics term remain unchanged under ordering, confirming that $\mathcal{F}$ is indeed monotonic.

Next, we show that $\mathcal{F}$ is continuous with respect to the pointwise supremum of an $\omega$-chain $\{d_n\}$. Consider the sequence $\{d_n\}$ forming an increasing chain in $\mathcal{M}$, meaning $d_n(z_i, z_j)$ converges to $d_\infty(z_i, z_j)$. By the properties of the Wasserstein metric and $L1$ norm, limits commute with these operations, leading to:

$$
\begin{aligned}
\mathcal{F}^\pi(\bigsqcup_{n \in \mathbb{N}} d_n)(z_i, z_j) = & |R_i^\pi(s_i^+, a_i) - R_j^\pi(s_j^+, a_j)| + W_2(\bigsqcup_{n \in \mathbb{N}} d_n)(T_i^\pi(s_i^+, a_i), T_j^\pi(s_j^+, a_j)) \\
& + \|I_i^\pi(s_i^+, s_{i+1}^+) - I_j^\pi(s_j^+, s_{j+1}^+)\|_1 \\
= & |R_i^\pi(s_i^+, a_i) - R_j^\pi(s_j^+, a_j)| + \sup_{n \in \mathbb{N}} W_2(d_n)(T_i^\pi(s_i^+, a_i), T_j^\pi(s_j^+, a_j)) \\
& + \|I_i^\pi(s_i^+, s_{i+1}^+) - I_j^\pi(s_j^+, s_{j+1}^+)\|_1 \\
= & \sup_{n \in \mathbb{N}} (|R_i^\pi(s_i^+, a_i) - R_j^\pi(s_j^+, a_j)| + W_2(d_n)(T_i^\pi(s_i^+, a_i), T_j^\pi(s_j^+, a_j)) \\
& + \|I_i^\pi(s_i^+, s_{i+1}^+) - I_j^\pi(s_j^+, s_{j+1}^+)\|_1) \\
= & (\bigsqcup_{n \in \mathbb{N}} \mathcal{F}^\pi(d_n))(z_i, z_j)
\end{aligned} \tag{15}
$$

This guarantees that $\mathcal{F}$ is continuous. The policy $\pi$ in the above equations is fixed. As the policy updates, it can converge to a fixed point.

Since $\mathcal{M}$ is a complete partial order ($\omega$-CPO) and $\mathcal{F}$ is a monotonic and continuous operator, the fixed-point theorem (Ferns et al., 2004) ensures the existence of a least fixed point $\tilde{d}$ such that $\mathcal{F}(\tilde{d}) = \tilde{d}$.

$\square$

**Theorem 1** (Value difference bound). *Given two tasks $\mathcal{M}_i$ and $\mathcal{M}_j$ in the latent space with states $s_i^+, s_j^+ \in \mathcal{S}^+$, and let $V^\pi$ be the value function of policy $\pi$, the value difference bound between the tasks can be given by:*

$$
|V^\pi(s_i^+) - V^\pi(s_j^+)| \leq d_\pi(z_i, z_j), \tag{16}
$$

*where $d_\pi(z_i, z_j)$ is the latent task belief metric.*

*Proof.* We will use the standard value function update:

$$V_n^\pi(s) = R^\pi(s,a) + \gamma \sum_{s' \in S} P^\pi(s' \mid s, a) V_{n-1}^\pi(s')$$

with $V_0^\pi(s) = 0$, and our task difference metric $d_\pi(z_i, z_j)$, and prove this by induction, showing that for all $n \in \mathbb{N}$ and states $s_i^+, s_j^+ \in \mathcal{S}^+$:

$$|V_n^\pi(s_i^+) - V_n^\pi(s_j^+)| \le d_\pi^n(z_i, z_j).$$

The base case holds trivially:

$$0 = V_0^\pi(s_i^+) - V_0^\pi(s_j^+) = 0, \quad \text{so assume this holds for } n.$$

Now, for $n+1$:

$$|V_{n+1}^\pi(s_i^+) - V_{n+1}^\pi(s_j^+)| = |R_i^\pi(s_i^+, a_i) + \gamma \sum_{s' \in S^+} T_i^\pi(s_i'^+ \mid s_i^+, a_i) V_n^\pi(s_i'^+) - R_j^\pi(s_j^+, a_j) - \gamma \sum_{s' \in S^+} T_j^\pi(s_j'^+ \mid s_j^+, a_j) V_n^\pi(s_j'^+)|.$$

We decompose this into rewards and transition dynamics:

$$\le |R_i^\pi(s_i^+, a_i) - R_j^\pi(s_j^+, a_j)| + \gamma \left| \sum_{s' \in S^+} \left( T_i^\pi(s_i'^+ \mid s_i^+, a_i) V_n^\pi(s_i'^+) - T_j^\pi(s_j'^+ \mid s_j^+, a_j) V_n^\pi(s_j'^+) \right) \right|.$$

Using the Lipschitz property of the value function, we can bound the transition dynamics difference:

$$\le |R_i^\pi(s_i^+, a_i) - R_j^\pi(s_j^+, a_j)| + \gamma W_2 \left( T_i^\pi(s_i^+, a_i), T_j^\pi(s_j^+, a_j) \right) + \left\| I_i^\pi(s_i^+, s_i'^+) - I_j^\pi(s_j^+, s_j'^+) \right\|_1.$$

By the definition of the latent task belief metric $d_\pi(z_i, z_j)$, we know:

$$d_\pi(z_i, z_j) = |R_i^\pi(s_i^+, a_i) - R_j^\pi(s_j^+, a_j)| + W_2 \left( T_i^\pi(s_i^+, a_i), T_j^\pi(s_j^+, a_j) \right) + \left\| I_i^\pi(s_i^+, s_i'^+) - I_j^\pi(s_j^+, s_j'^+) \right\|_1.$$

We assume that $z$ contains sufficient information about the task differences at the current time step, and we disregard the effect of the discount factor $\gamma$ during the experiments. Thus, we have:

$$|V_{n+1}^\pi(s_i^+) - V_{n+1}^\pi(s_j^+)| \le d_\pi^{n+1}(z_i, z_j).$$

$\square$

**Lemma 2.** *Let $g$ be an $(\epsilon_R, \epsilon_T, \epsilon_I)$-approximate bisimulation abstraction of $M$. The states in the real space are mapped to the latent dynamics space $\mathcal{S}^+$ through $g$. For any two states in the real space, $s_1$ and $s_2$, if their representations in the latent space are identical, i.e., $g(s_1) = g(s_2)$, then*

$$|R(s_1, a) - R(s_2, a)| \le \epsilon_R,$$

$$\|T(s_1, a) - T(s_2, a)\|_1 \le \epsilon_T,$$

$$\|I(s_1, s_1') - I(s_2, s_2')\|_1 \le \epsilon_I.$$

*Proof.* The proof follows a similar process to Lemma 3 in (Jiang, 2018). Here, we have added a representation constraint on the inverse dynamics. $\square$

**Theorem 2** (Latent transfer bound). *Let $Q_{\mathcal{M}_j}^*$ be the optimal Q-function for task $\mathcal{M}_j$. The difference between $Q_{\mathcal{M}_j}^*$ and the Q-function of the policy $\pi$ learned from task $\mathcal{M}_i$, applied to task $\mathcal{M}_j$, is bounded as follows:*

$$\left\| Q_{\mathcal{M}_j}^* - [Q_{\mathcal{M}_i}^\pi]_{\mathcal{M}_j} \right\|_\infty \le \epsilon_R + \gamma \left( \epsilon_T + \epsilon_I + \|z_i - z_j\|_1 \right) \frac{R_{max}}{2(1-\gamma)}. \tag{17}$$

*Proof.* $\|z_i - z_j\|_1$ is the distance between the latent task beliefs of task $\mathcal{M}_i$ and task $\mathcal{M}_j$. In the real state space, the augmented state is still conditioned on $z_l$ (Section 3.3).

We aim to bound the difference between the Q-function $Q^*_{\mathcal{M}_j}$ (the optimal Q-function for task $\mathcal{M}_j$) and $[Q^\pi_{\mathcal{M}_i}]_{\mathcal{M}_j}$ (the Q-function for policy $\pi$ learned on task $\mathcal{M}_i$, but applied to task $\mathcal{M}_j$). This difference is given by:

$$\|Q^*_{\mathcal{M}_j} - [Q^\pi_{\mathcal{M}_i}]_{\mathcal{M}_j}\|_\infty.$$

The Q-function for any task satisfies the Bellman equation:

$$Q(s,a) = R(s,a) + \gamma \mathbb{E}_{s' \sim T(s,a)} \left[ \max_{a'} Q(s',a') \right].$$

Using this, we decompose the Q-function difference:

$$\|Q^*_{\mathcal{M}_j} - [Q^\pi_{\mathcal{M}_i}]_{\mathcal{M}_j}\|_\infty \leq \frac{1}{1-\gamma} \|Q^*_{\mathcal{M}_j} - \mathcal{T}^\pi_{\mathcal{M}_j} Q^\pi_{\mathcal{M}_i}\|_\infty.$$

Here, $\mathcal{T}^\pi_{\mathcal{M}_j} Q^\pi_{\mathcal{M}_i} = R_{\mathcal{M}_j}(s^+, a) + \gamma \mathbb{E}_{s' \sim T_{M_j}(s^+, a), a \sim I_{M_j}(s^+, s'^+)}[Q^\pi_{\mathcal{M}_i}(s'^+, a')]$.

According to Lemma 2,

$$\sup_{s_i^+, s_j^+ \in \mathcal{S}^+, g(s_i)=g(s_j)} \left\| T_{\mathcal{M}_i}(s_i^+, a) - T_{\mathcal{M}_j}(s_j^+, a) \right\|_1$$

$$\leq \sup_{s_i^+, s_j^+ \in \mathcal{S}^+, g(s_i)=g(s_j)} \left( \left\| T_{\mathcal{M}_i}(s_i^+, a) - T_{\mathcal{M}_i}(s_j^+, a) \right\|_1 + \left\| T_{\mathcal{M}_i}(s_j^+, a) - T_{\mathcal{M}_j}(s_j^+, a) \right\|_1 \right)$$

$$\leq \sup_{s_i^+, s_j^+ \in \mathcal{S}^+, g(s_i)=g(s_j)} \left\| T_{\mathcal{M}_i}(s_i^+, a) - T_{\mathcal{M}_i}(s_j^+, a) \right\|_1 + \sup_{s_i^+, s_j^+ \in \mathcal{S}^+, g(s_i)=g(s_j)} \left\| T_{\mathcal{M}_i}(s_j^+, a) - T_{\mathcal{M}_j}(s_j^+, a) \right\|_1$$

$$= \epsilon_T + \left\| T_{\mathcal{M}_i}(s_j^+, a) - T_{\mathcal{M}_j}(s_j^+, a) \right\|_1, \tag{18}$$

$$\sup_{s_i^+, s_j^+ \in \mathcal{S}^+, g(s_i)=g(s_j)} |R_{\mathcal{M}_i}(s_i^+, a) - R_{\mathcal{M}_j}(s_j^+, a)|$$

$$\leq \sup_{s_i^+, s_j^+ \in \mathcal{S}^+, g(s_i)=g(s_j)} \left( |R_{\mathcal{M}_i}(s_i^+, a) - R_{\mathcal{M}_i}(s_j^+, a)| + |R_{\mathcal{M}_i}(s_j^+, a) - R_{\mathcal{M}_j}(s_j^+, a)| \right)$$

$$\leq \sup_{s_i^+, s_j^+ \in \mathcal{S}^+, g(s_i)=g(s_j)} |R_{\mathcal{M}_i}(s_i^+, a) - R_{\mathcal{M}_i}(s_j^+, a)| + \sup_{s_i^+, s_j^+ \in \mathcal{S}^+, g(s_i)=g(s_j)} |R_{\mathcal{M}_i}(s_j^+, a) - R_{\mathcal{M}_j}(s_j^+, a)|$$

$$= \epsilon_R + |R_{\mathcal{M}_i}(s_j^+, a) - R_{\mathcal{M}_j}(s_j^+, a)|, \tag{19}$$

$$\sup_{s_i^+, s_j^+ \in \mathcal{S}^+, g(s_i)=g(s_j)} \left\| I_{\mathcal{M}_i}(s_i^+, s_i'^+) - I_{\mathcal{M}_j}(s_j^+, s_j'^+) \right\|_1$$

$$\leq \sup_{s_i^+, s_j^+ \in \mathcal{S}^+, g(s_i)=g(s_j)} \left( \left\| I_{\mathcal{M}_i}(s_i^+, s_i'^+) - I_{\mathcal{M}_i}(s_j^+, s_j'^+) \right\|_1 + \left\| I_{\mathcal{M}_i}(s_j^+, s_j'^+) - I_{\mathcal{M}_j}(s_j^+, s_j'^+) \right\|_1 \right)$$

$$\leq \sup_{s_i^+, s_j^+ \in \mathcal{S}^+, g(s_i)=g(s_j)} \left\| I_{\mathcal{M}_i}(s_i^+, s_i'^+) - I_{\mathcal{M}_i}(s_j^+, s_j'^+) \right\|_1 + \sup_{s_i^+, s_j^+ \in \mathcal{S}^+, g(s_i)=g(s_j)} \left\| I_{\mathcal{M}_i}(s_j^+, s_j'^+) - I_{\mathcal{M}_j}(s_j^+, s_j'^+) \right\|_1$$

$$= \epsilon_I + \left\| I_{\mathcal{M}_i}(s_j^+, s_j'^+) - I_{\mathcal{M}_j}(s_j^+, s_j'^+) \right\|_1, \tag{20}$$

$$\left\| T_{\mathcal{M}_i}(s_i^+, a) - T_{\mathcal{M}_j}(s_j^+, a) \right\|_1 + |R_{\mathcal{M}_i}(s_i^+, a) - R_{\mathcal{M}_j}(s_j^+, a)| + \left\| I_{\mathcal{M}_i}(s_i^+, s_i'^+) - I_{\mathcal{M}_j}(s_j^+, s_j'^+) \right\|_1$$

$$\leq \epsilon_R + \epsilon_T + \epsilon_I + \|z_i - z_j\|_1, \tag{21}$$

$$\|Q^*_{\mathcal{M}_j} - \mathcal{T}^\pi_{\mathcal{M}_j} Q^\pi_{M_i}\|_\infty$$

$$= \left| (\mathcal{T}_{\mathcal{M}_j} Q^*_{\mathcal{M}_j})(s'^+_j, a) - (\mathcal{T}[Q^*_{\mathcal{M}_j}]_{\mathcal{M}_i})(s'^+_i, a) \right|$$

$$= \left| R_{\mathcal{M}_j}(s'^+_j, a) + \gamma \langle T_{\mathcal{M}_j}(s'^+_j, a), V^*_{\mathcal{M}_j} \rangle - R_{\mathcal{M}_i}(s'^+_i, a) - \gamma \langle T_{\mathcal{M}_i}(s'^+_i, a), [V^*_{\mathcal{M}_j}]_{\mathcal{M}_i} \rangle \right|$$

$$\leq \epsilon_R + \gamma \left| \langle T_{\mathcal{M}_j}(s'^+_j, a), V^*_{\mathcal{M}_j} \rangle - \langle T_{\mathcal{M}_i}(s'^+_i, a), [V^*_{\mathcal{M}_j}]_{\mathcal{M}_i} \rangle \right|$$

$$\leq \epsilon_R + \gamma \left| \langle T_{\mathcal{M}_j}(s'^+_j, a), V^*_{\mathcal{M}_j} \rangle - \langle T_{\mathcal{M}_i}(s'^+_i, a), [V^*_{\mathcal{M}_j}]_{\mathcal{M}_i} \rangle \right| + \gamma |I_{\mathcal{M}_i}(s^+_j, s'^+_j) - I_{\mathcal{M}_j}(s^+_j, s'^+_j)|$$

$$\leq \epsilon_R + \gamma(\epsilon_T + \epsilon_I + \|z_i - z_j\|_1)\|V^*_{\mathcal{M}_j} - \frac{R_{\max}}{2(1-\gamma)}\mathbf{1}\|_\infty$$

$$\leq \epsilon_R + \gamma(\epsilon_T + \epsilon_I + \|z_i - z_j\|_1)\frac{R_{\max}}{2(1-\gamma)}.$$

We get the final bound:

$$\left\| Q^*_{\mathcal{M}_j} - [Q^\pi_{\mathcal{M}_i}]_{\mathcal{M}_j} \right\|_\infty \leq \epsilon_R + \gamma \left( \epsilon_T + \epsilon_I + \|z_i - z_j\|_1 \right) \frac{R_{\max}}{2(1-\gamma)}.$$

$\square$

## C  SIMBELIEF PSEUDO-CODE

---
**Algorithm 1** SimBelief algorithm
---
**Input**: Task distribution $p(M)$
**Initialise**: context encoder $q_\phi$, real envs dynamics $p_\phi$, actor $\pi_\theta$, critic $Q_\omega$, replay buffer $\mathcal{D}$, belief similarity learner $\psi_l$, latent dynamics $p_\theta$, state encoder $g$, distribution offset $(\Delta\mu, \Delta\sigma)$
 1: **while** not done **do**
 2:      Sample tasks $M_{train} = \{\mathcal{M}_i\}_{i=1}^N$ from $p(M)$
 3:      Collect trajectories with $\pi_\theta$ and add to buffer $\mathcal{D}$                 ▷ Data collection
 4:      **for** step in SAC training steps **do**                       ▷ Training step
 5:          Sample training tasks $M_{\text{train}}$ from $p(M)$
 6:          Sample batches from $\mathcal{D}$
 7:          Infer specific task beliefs $z_r \sim \psi_r(b_r \mid h)q_\phi(h \mid \tau_{:t})$
 8:          Infer latent task beliefs $z_l \sim \psi_l(b_l \mid h)q_\phi(h \mid \tau_{:t})$
 9:          Permute $z_l^i$ to get $z_l^j$
10:          $s_i^+ = (g(s), z_i), s_j^+ = (g(s), z_j)$
11:          Update latent dynamics $p_\theta$ and state encoder $g$ using Eq. 4
12:          Update belief similarity learner $\psi_l$ using Eq. 8
13:          Update distribution offset $(\Delta\mu, \Delta\sigma)$ using Eq. 9
14:          Integrate $b_l^i$ with $b_r^i$ to obtain $b^i$
15:          Update $(\theta, \omega)$ with SAC algorithm                   ▷ SAC update
16:      **end for**
17:      **for** step in VAE training steps **do**
18:          Sample $\tau_{:T} \sim \mathcal{B}$ with trajectory length $T$
19:          Decode only the past trajectories and update $q_\phi$, $\psi_r$ and $p_\phi$ using Eq. 6    ▷ VAE update
20:      **end for**
21: **end while**
---

## D  CONTEXT-BASED META-RL BASELINES

In this section, we provide a detailed overview of the baseline methods compared in our experiments, highlighting their core methodologies and design principles.

**PEARL** (Rakelly et al., 2019) (Probabilistic Embeddings for Actor-Critic RL) is an off-policy meta-reinforcement learning algorithm designed to enhance both meta-training sample efficiency and

rapid adaptation to new tasks. By disentangling task inference from control, PEARL employs a probabilistic latent context variable that enables structured exploration and efficient posterior sampling. This design allows the policy to reason about task uncertainty and adapt quickly in sparse reward or dynamic environments. Built upon the Soft Actor-Critic framework, PEARL achieves better sample efficiency compared to on-policy meta-RL methods.

**MetaCURE** (Zhang et al., 2021c) (Meta-RL with Efficient Uncertainty Reduction Exploration) is an off-policy meta-RL framework designed to address the challenges of sparse-reward environments. It explicitly separates exploration and exploitation by learning distinct policies for each, enhancing the efficiency of task inference and adaptation. The exploration policy is driven by an empowerment-based intrinsic reward that maximizes information gain about the task, enabling efficient collection of task-relevant experiences. MetaCURE leverages a shared probabilistic task inference mechanism, which improves sample efficiency by integrating exploration and exploitation processes. Compared to MetaCURE, SimBelief does not require knowledge of task IDs during the training phase. Instead, it learns the common structure of tasks to enable reasoning, resulting in stronger online adaptation capabilities.

**VariBAD** (Zintgraf et al., 2019) (Variational Bayes-Adaptive Deep RL) is a meta-reinforcement learning framework that approximates Bayes-optimal policies by leveraging variational inference and latent task embeddings. The algorithm learns a posterior belief over tasks using a variational auto-encoder and conditions its policy on this belief to balance exploration and exploitation in uncertain environments. VariBAD is notable for its ability to perform structured online exploration by integrating task uncertainty directly into action selection. Unlike traditional methods that rely on computationally intractable posterior sampling or explicit planning, VariBAD offers a tractable and flexible approach to Bayes-adaptive policies.

**HyperX** (Zintgraf et al., 2021) (Hyper-State Exploration) is a meta-reinforcement learning method that addresses sparse reward environments by leveraging exploration bonuses to meta-learn approximately Bayes-optimal task-adaptation strategies. It integrates two exploration bonuses during meta-training: (1) a bonus based on hyper-states (combining environment states and task beliefs) to encourage diverse task-exploration strategies, and (2) a reconstruction error bonus to incentivize the agent to collect data where task beliefs are inaccurate. By exploring hyper-states, HyperX efficiently gathers data for belief inference and optimally trades off exploration and exploitation in sparse or complex environments.

**RL$^2$** (Duan et al., 2016) reformulates the reinforcement learning process as a meta-learning problem, embedding task-specific learning within the hidden state of a recurrent neural network (RNN). By leveraging a "slow" reinforcement learning algorithm to optimize the RNN's weights, RL$^2$ enables the network to store and process information about task dynamics across episodes. This design allows the agent to efficiently adapt to unseen tasks by utilizing its historical trajectory data, including observations, actions, rewards, and termination flags. RL$^2$ demonstrates competitive performance in solving multi-armed bandits and tabular MDPs, achieving results comparable to theoretically optimal algorithms. Additionally, its scalability to high-dimensional tasks, such as visual navigation in dynamic environments, highlights its potential as a versatile and efficient meta-RL framework.

## E  ENVIRONMENTS AND IMPLEMENTATION DETAILS

Table 1: Adaptation length and goal settings for environments used for evaluation

| Environment | # of adaptation episodes | Max steps per episode | Goal type | Goal range | Goal radius |
|---|---|---|---|---|---|
| Cheetah-Vel-Sparse | 2 | 200 | Velocity | [0,3] | 0.5 |
| Point-Robot-Sparse | 2 | 60 | Position | Semicircle with radius 1 | 0.3 |
| Walker-Rand-Params | 2 | 200 | Velocity | 1.5 | 0.5 |
| Panda-Reach | 3 | 50 | Position | / | 0.05 |
| Panda-Push | 3 | 50 | Position | / | 0.05 |
| Panda-Pick-And-Place | 3 | 50 | Position | / | 0.05 |

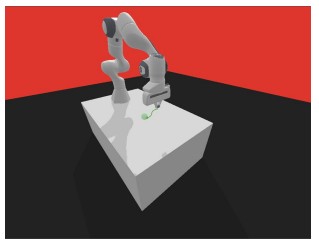 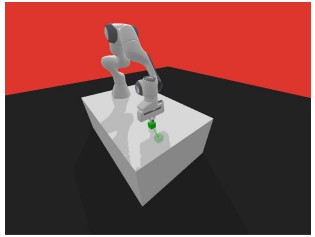 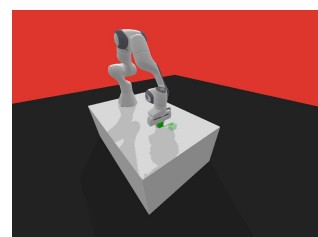

**Panda-Reach**      **Panda-Push**      **Panda-Pick-And-Place**

Figure 7: Panda-gym tasks. The Panda-Reach task involves controlling the Panda robotic arm to move its gripper to a randomly generated target position within a defined workspace, the Panda-Push task requires pushing a cube from its initial position to a randomly generated target position on a table, and the Panda-Pick-And-Place task focuses on picking up a cube and placing it at a randomly generated target location above the table.

Table 1 summarizes key settings for the environments used in this work, including the number of adaptation episodes, the maximum number of steps allowed per episode, the type of goal (whether velocity or position-based), the goal range, and the goal radius. The specific environments are described as follows:

**Cheetah-Vel-Sparse.** The Cheetah-Vel-Sparse environment involves controlling a half-cheetah robot to achieve and maintain a target velocity, which is randomly sampled from a uniform distribution over the range [0, 3]. At each time step, the robot receives a reward based on how closely its current velocity matches the target velocity, with a sparse reward system in place. Specifically, if the difference between the current velocity and the target velocity is within a specified tolerance (goal radius of 0.5), the robot receives a reward; otherwise, the reward is zero. Additionally, a control cost is applied to penalize large actions. The observation space includes the robot's joint positions, velocities, and body orientation, while the action space consists of motor torques controlling the robot's movements. The sparse reward encourages the agent to achieve the target velocity while minimizing control effort.

**Point-Robot-Sparse.** The Point-Robot-Sparse environment requires a point robot to navigate towards a goal randomly placed along a unit half-circle. The goal position is sampled at the beginning of each episode, and the robot receives rewards only when it reaches within a goal radius of 0.3 units from the target. The observation space consists of the robot's current position, and the action space allows movement in both x and y directions. The reward system is sparse, meaning that rewards are only given when the robot is within the specified goal radius, encouraging the robot to explore and move efficiently towards the target. The sparse reward structure makes the task more challenging, as the robot receives feedback only when it approaches the target, requiring it to learn how to navigate effectively with limited guidance.

**Walker-Rand-Params.** The Walker-Rand-Params environment involves controlling a bipedal walker robot, with the added complexity of randomized physical parameters such as body mass, leg strength, and joint properties. These parameters are randomized at the start of each episode, requiring the robot to adapt to various physical configurations in order to move forward. The observation space includes the walker's joint angles, positions, and velocities, while the action space consists of motor commands applied to the walker's joints. The reward system is primarily based on how closely the walker's velocity matches a target velocity of 1.5 units per time step. If the deviation from the target velocity exceeds 0.5 units, the reward is set to 0. For smaller deviations, the reward is given as 0.8 - distance from the target velocity. Additionally, a small control cost proportional to the sum of the squared motor actions is subtracted from the reward to penalize large actions.

**Panda-Reach.** The Panda-Reach task involves controlling the Panda robotic arm's gripper to reach a target position, randomly generated within a 30cm × 30cm × 30cm workspace. The task is considered successful when the distance between the gripper and the target is less than 5 cm. The observation space for this task includes the position and velocity of the gripper (6 coordinates), augmented by two additional vectors representing the desired goal (target position) and the achieved goal (current gripper position). The action space comprises three movement coordinates (x, y, z) for controlling the gripper. The gripper remains closed throughout the task, and its movement along

these axes determines the success of the task. A sparse reward function is used, with a reward of 0 if the target is reached, and -1 otherwise.

**Panda-Push.** The Panda-Push task requires the robot to push a cube (side length of 4 cm) placed on a table to a target position. Both the target position and the initial position of the cube are randomly generated within a $20\text{cm} \times 20\text{cm}$ area around the neutral position of the robot. The gripper remains closed during the task, and the objective is to move the cube to within 5 cm of the target position. The observation space includes the gripper's position and velocity (6 coordinates), along with the cube's position, orientation, and velocity (12 coordinates). The action space consists of three movement coordinates (x, y, z) for controlling the gripper's movement. As in the Panda-Reach task, a sparse reward function is used, providing a reward of 0 if the task is completed successfully, and -1 otherwise.

**Panda-Pick-And-Place.** The Panda-Pick-And-Place task involves the robot picking up a cube (side length of 4 cm) and placing it at a target location, which is randomly generated within a $20\text{cm} \times 20\text{cm} \times 10\text{cm}$ volume above the table. The task is completed when the cube is placed within 5 cm of the target position. The observation space for this task includes the gripper's position and velocity (6 coordinates), the cube's position, orientation, and velocity (12 coordinates), and the opening state of the gripper (1 coordinate). The action space is expanded to include three movement coordinates (x, y, z) for controlling the gripper and an additional coordinate for controlling the gripper's opening and closing. A sparse reward function is employed, rewarding the robot with 0 for completing the task and -1 otherwise (Figure 7).

All experiments were conducted using an Nvidia RTX 4090 GPU, the source code is available at: `https://github.com/mlzhang-pr/SimBelief`.

Table 2: Hyperparameter settings for SimBelief in different environments

| Parameter Name | Cheetah-Vel-Sparse Walker-Rand-Params | Point-Robot-Sparse | Panda-Reach | Panda-Push Panda-Pick-And-Place |
|---|---|---|---|---|
| Number of Tasks | 120 | 100 | 100 | 60 |
| Number of Training Tasks | 100 | 80 | 80 | 50 |
| Number of Evaluation Tasks | 20 | 20 | 20 | 10 |
| Number of Episodes | 2 | 2 | 3 | 3 |
| Number of Iterations | 1000 | 2000 | 1000 | 4000 |
| RL Updates per Iteration | 2000 | 1000 | 1000 | 1000 |
| Batch Size | 256 | 256 | 256 | 256 |
| Policy Buffer Size | 1e6 | 1e6 | 1e6 | 1e6 |
| VAE Buffer Size | 1e5 | 5e4 | 5e4 | 5e4 |
| Policy Layers | [128, 128, 128] | [128, 128] | [128, 128] | [128, 128, 128] |
| Actor Learning Rate | 0.0003 | 0.00007 | 0.00007 | 0.00007 |
| Critic Learning Rate | 0.0003 | 0.00007 | 0.00007 | 0.00007 |
| Discount Factor ($\gamma$) | 0.99 | 0.9 | 0.9 | 0.9 |
| Entropy Alpha | 0.2 | 0.01 | 0.01 | 0.01 |
| VAE Updates per Iteration | 20 | 25 | 25 | 25 |
| VAE Learning Rate | 0.0003 | 0.001 | 0.001 | 0.001 |
| KL Weight | 1.0 | 0.1 | 0.1 | 0.1 |
| Task Embedding Size | 10 | 10 | 5 | 5 |

# F  THE MECHANISM OF SIMBELIEF FOR RAPID ADAPTATION

In this section, we will explain, from the perspective of task belief representation, why SimBelief can quickly converge to a near Bayes-optimal policy during the training phase and how it enables adaptation to OOD tasks within a single episode in sparse reward environments.

## F.1  THE EVOLUTION OF TASK BELIEF DURING THE TRAINING PHASE

During the training phase, the agent needs to explore complex environments to acquire relevant information about the current task, which is represented by the specific task belief $b_r$. Other meta-RL algorithms are limited to task-specific information while ignoring the similarities between tasks. SimBelief's latent dynamics can capture the unique structure shared among tasks and distinguish these structures through the latent task belief $b_l$. Specifically, during the learning process, SimBelief

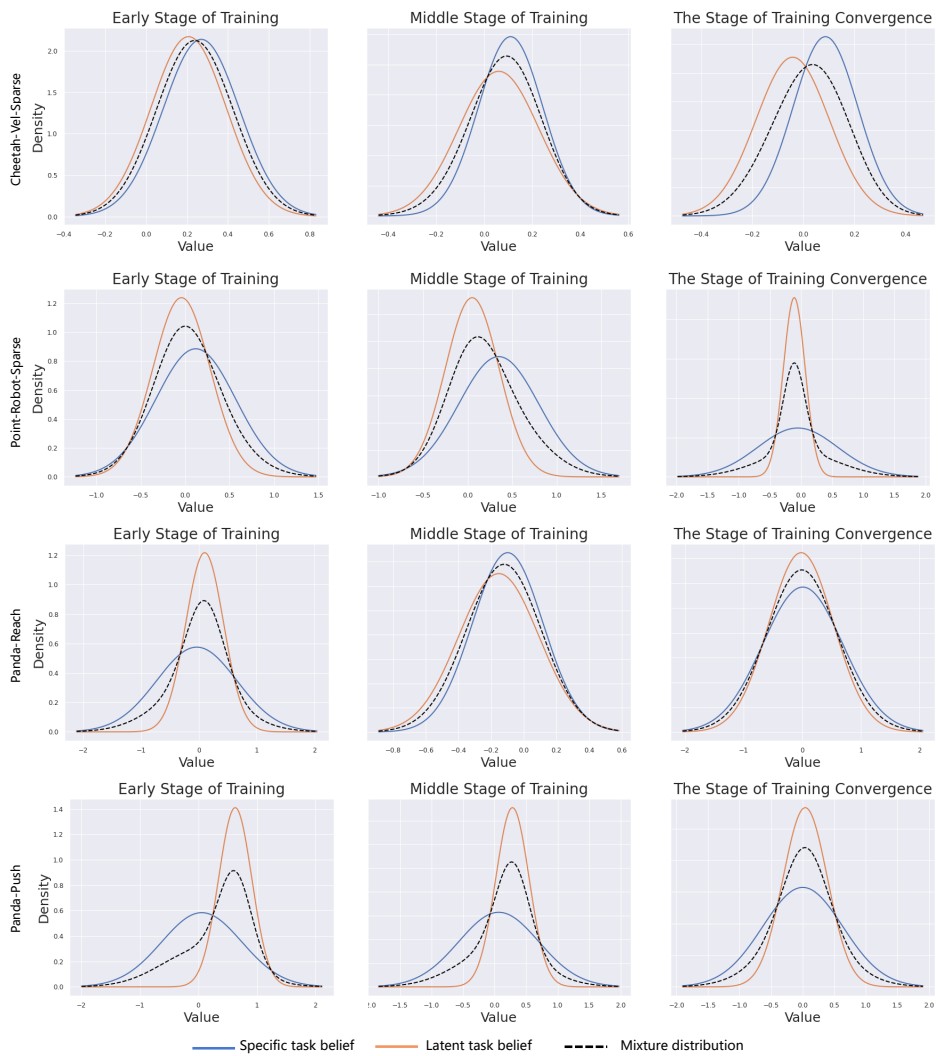

Figure 8: Visualization of different task beliefs during the training phase.

identifies the distribution of similar tasks via $b_l$, and then uses $b_r$ to fit the agent to the exact specific task distribution, as illustrated in Figure 1. In Figure 8, we visualize the specific task belief, latent task belief, and their mixed Gaussian distribution. During the early and middle stages of training, the agent quickly identifies the distribution of the task being executed. After convergence, it can accurately fit the specific task distribution.

## F.2 THE CORRELATION OF BELIEFS BETWEEN DIFFERENT TASKS DURING THE ADAPTATION PHASE

We visualize the correlations of task beliefs across different tasks in Point-Robot-Sparse, Panda-Reach, Panda-Push, and Panda-Pick-Place environments. In each environment, 20 tasks are randomly generated, and the SimBelief agent's rollouts are used to extract the specific task beliefs and latent task beliefs. The cosine similarity between task beliefs across tasks is calculated, and the correlation matrices are visualized. As shown in Figure 9, specific task belief primarily captures the local information of task distributions (learning weaker correlations between tasks), while latent task belief emphasizes global information (capturing stronger inter-task correlations). This illustrates the principle behind SimBelief's ability to achieve rapid adaptation to OOD tasks in sparse reward environments: *latent task belief enhances the agent's reasoning efficiency across tasks and*

*improves the efficiency of knowledge transfer.* Theorem 2 demonstrates the transferability of tasks in the latent space.

## G  ABLATION STUDY

### G.1  LATENT DYNAMICS ABLATION

To validate the effectiveness of our algorithm using the learned latent task belief, we froze the overall latent dynamics and, similar to VariBAD (Zintgraf et al., 2019), only used an external VAE for environment reconstruction. Our algorithm demonstrated significant improvements across all six tested tasks, with particularly notable gains in the more exploration-demanding tasks, Panda-Push and Panda-Pick-And-Place (Figure 10).

### G.2  THE IMPACT OF INVERSE DYNAMICS ON ADAPTATION ABILITY

To demonstrate the effectiveness of the inverse dynamics module in facilitating rapid learning of task similarity, we conducted ablation experiments by removing the inverse dynamics module from the latent space on Cheetah-Vel-Sparse (velocity range [4.0,5.0]) and Point-Robot-Sparse (radius=1.2) tasks. The results show that the inverse dynamics module plays a critical role in enabling the agent's reasoning and generalization on OOD tasks (Figure 11).

### G.3  ABLATION STUDY ON $w_r$ AND $w_l$

When combining $b_l$ and $b_r$, we apply a mixture of Gaussians to the two beliefs. To evaluate the impact of different weights, we compared three weighting configurations on the Cheetah-Vel-Sparse task: $[w_r, w_l] = [0.5, 0.5]$, $[w_r, w_l] = [0.25, 0.75]$, and $[w_r, w_l] = [0.75, 0.25]$. Among these, $[w_r, w_l] = [0.5, 0.5]$ achieved higher exploration efficiency during training and demonstrated stronger adaptability to OOD tasks (Figure 12).

### G.4  ABLATION STUDY ON OFFSET $(\Delta\mu, \Delta\sigma)$

During the training phase, we use the offset $(\Delta\mu, \Delta\sigma)$ applied to the latent task belief primarily to improve the stability of the algorithm's convergence in continuous control tasks (e.g., Cheetah-Vel-Sparse). However, this does not have a significant impact on the overall performance of the algorithm. Even without using the offset during training, SimBelief still outperforms other baselines (Figure 13).

## H  ADDITIONAL VISUALIZATIONS

In this section, we provide the task belief representations of SimBelief and other baselines on the Walker-Rand-Param tasks (Figure 14), the adaptation performance on in-distribution tasks Panda-Pick-And-Place and Walker-Rand-Params (Figure 15), and the exploration performance on the in-distribution Point-Robot-Sparse task (Figure 16).

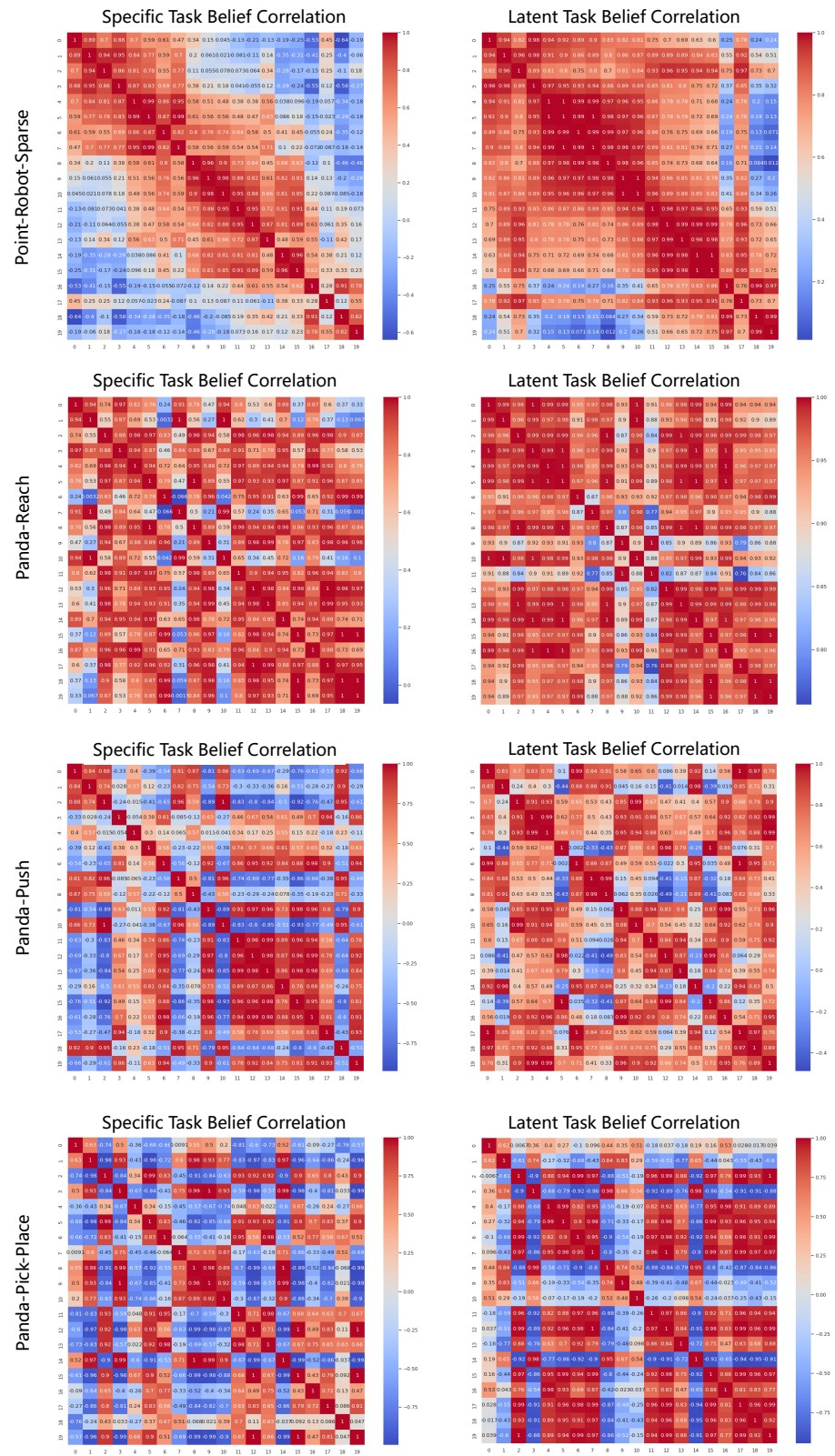

Figure 9: The correlation matrix of task beliefs across different tasks. Specific task belief focuses more on local information between tasks, while latent task belief captures the global characteristics of task distributions.

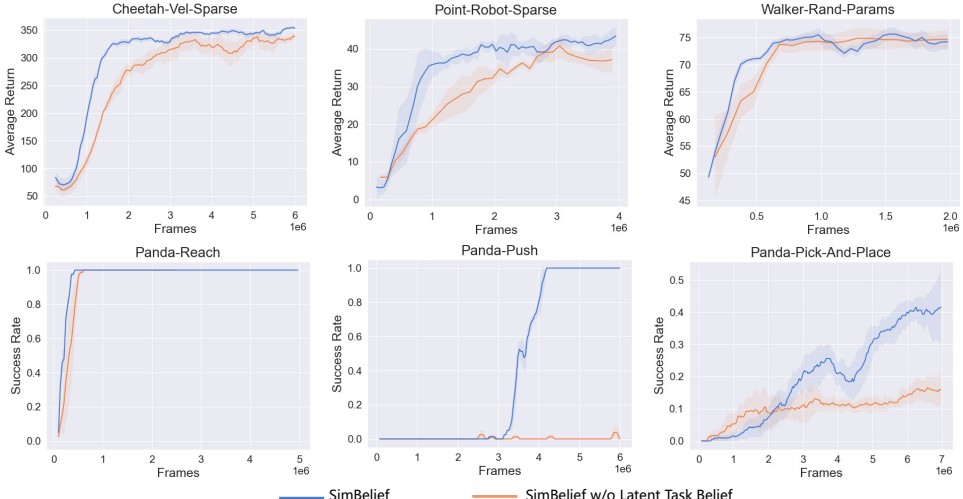

Figure 10: Ablation study on SimBlief's latent task belief.

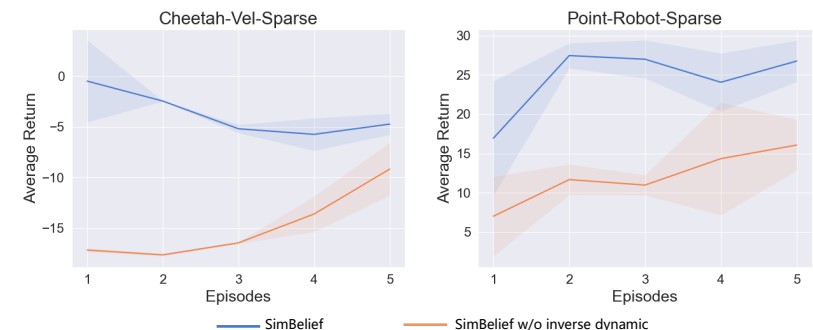

Figure 11: Ablation study on SimBlief's inverse dynamic module in latent space.

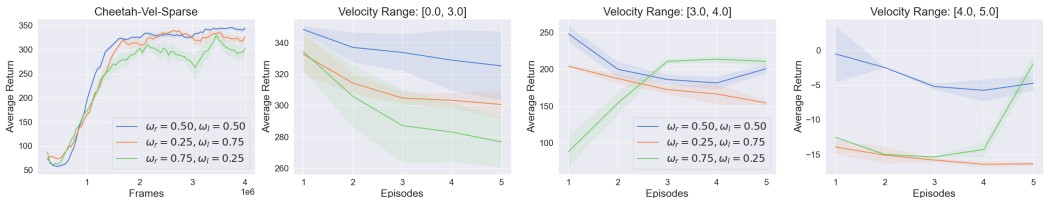

Figure 12: The performance of different Gaussian mixture weights on the Cheetah-Vel-Sparse task.

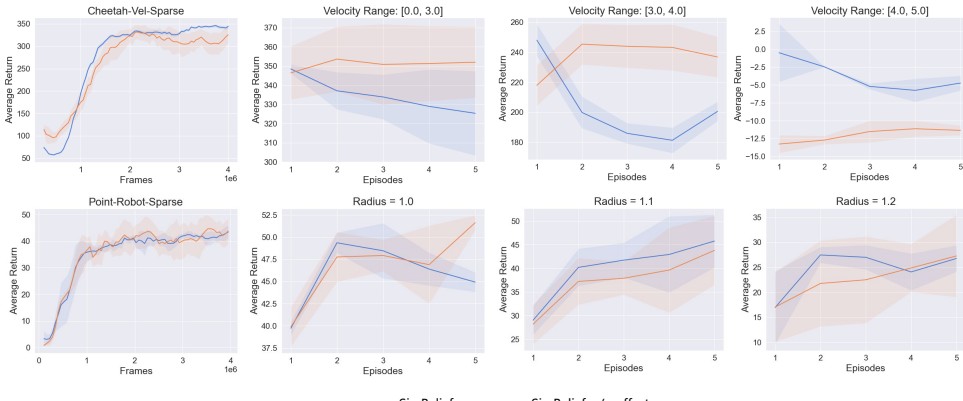

Figure 13: Ablation study on offset $(\Delta\mu, \Delta\sigma)$.

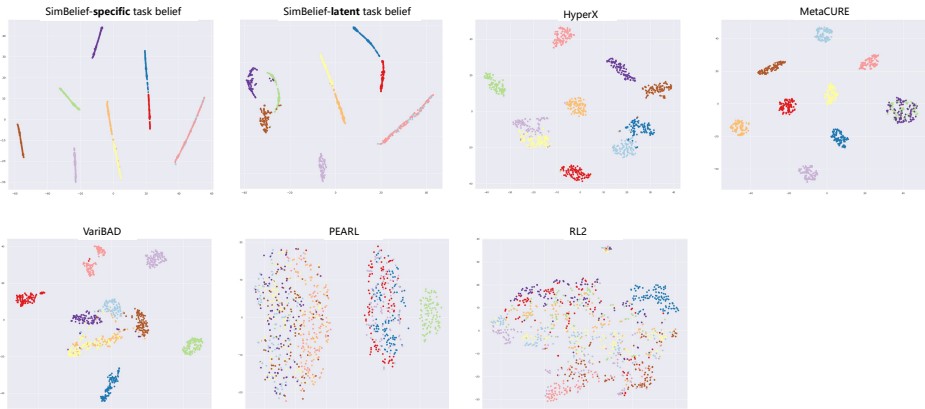

Figure 14: The task belief representation of all experimental algorithms for 10 random in-distribution Walker-Rand-Param tasks.

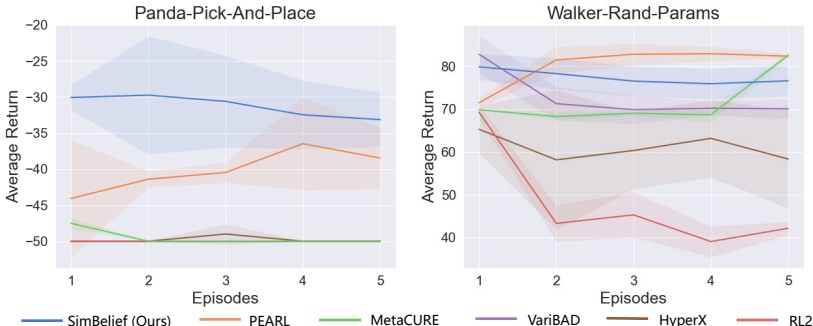

Figure 15: Average test performance for the first 5 rollouts on randomly generated tasks of Panda-Pick-And-Place and Walker-Rand-Params.

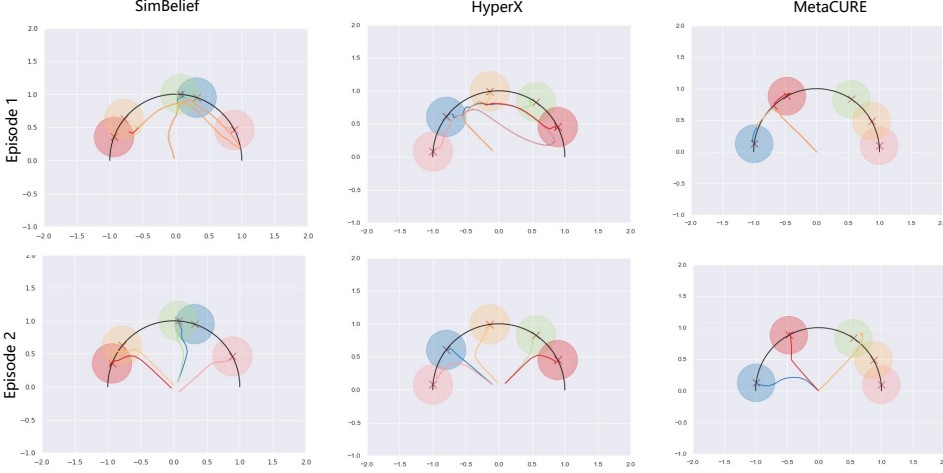

Figure 16: Exploration and adaptation performance in the 5 random in-distribution (radius = 1.0) tasks of Point-Robot-Sparse.

