# OpenReview forum: "Learning Task Belief Similarity with Latent Dynamics for Meta-Reinforcement Learning"
_ICLR.cc/2025/Conference — ICLR 2025 Poster_

### Official Review · Reviewer_z5au · 2024-10-24

**Soundness:** 3
**Presentation:** 3
**Contribution:** 3
**Rating:** 6
**Confidence:** 4

**Summary:**

This work considers meta RL, especially these sparse reward settings. This work proposes to utilize bisimulation metrics to extract behavioral similarity in continuous MDPs and proposes SimBelief, which can extract common features of similar task distributions. This work further theoretically validates the effectiveness of the latent task belief metric in BAMDPs. Experiments show the effectiveness of SimBelief, especially in sparse reward settings and o.o.d settings.

**Strengths:**

- It is novel to include Bisimulation to measure the task similarity.

- Sparse reward is a hot topic in the meta RL community.

- Several experiments show that SimBelief performs better than several baselines, especially in sparse reward settings.

**Weaknesses:**

- The writings of the proofs are poor with several typos.

(1) in lines 714-715, the value function update should be $V_{n+1} = ... V_n$

(2) in lines 719-710, $d^n$ here seems to be $d$

(3) in lines 733-739, why we can use the Lipschitz property of the value function to get the result? It seems the core to prove the result but the proof does not include a detailed discussion of this inequality.

(4) If you use the Lipschitz property of the value function, what is the Lipschitz constant?

(5) The proof of Theorem 2 is even more poor and needs to be thoroughly polished.

- The baselines compared in the paper are somehow old. The latest baseline in this paper is MetaCURE and  HyperX, which were published in 2021. There are several much more new baselines for meta RL, especially sparse reward settings or o.o.d. settings like [1-3], which should be discussed and compared. Also, it seems the evaluation environments are designed by this paper, are these previous works utilizing these environments?

- Sparse reward is indeed an important setting in RL, but why the proposed method can boost the agent's performance in sparse reward environments? As the reward structure may be independent of the dynamic structure, "extracting common features of similar task distributions" may not benefit the extremely sparse reward settings. Assume that there are n tasks that own the same dynamic, and we can only get the reward in the terminal step, there is no way to identify what is the current task as it is independent of the dynamic structure / previous rewards. Although it is not my major concern (experiments show that the proposed method can perform better than baselines in sparse settings), it will make the paper more solid if there are theoretical analyses about why the proposed method can handle sparse reward settings.

Ref:

[1] Learning Action Translator for Meta Reinforcement Learning on Sparse-Reward Tasks

[2] Meta-Reinforcement Learning Based on Self-Supervised Task Representation Learning

[3] Enhanced Meta Reinforcement Learning using Demonstrations in Sparse Reward Environments


------

**After rebuttal, most of my concerns have been addressed, especially about the theoretical proofs, and I have raised my score to 6.**

**Questions:**

See weaknesses.

---

> ### Author Response · Authors · 2024-11-25
>
> Thank you for your thorough and insightful review. We appreciate your feedback and suggestions, which have greatly helped us improve the quality of our work. Below, we address each of your comments and provide detailed responses.
>
> 1. **Writing of proofs and typos**
>
> We have revised the proofs to improve clarity and correctness, addressing typographical errors where the value function update should indeed be $V_n^\pi(s) = R^\pi(s, a) + \gamma \sum_{s' \in S} P^\pi(s' \mid s, a) V_{n-1}^\pi(s')$​.
>
> 2. **In lines 719-720 $d^{n}$  here seems to be $d$**
>
> Including $n$ emphasizes that the metric $d^{n}$ evolves over iterations, reflecting how the differences in dynamics propagate through the value function updates. If the task dynamics differences stabilize after a certain number of steps, $d^{n}$​ will converge, aligning with the final task similarity metric.
>
> 3. **Why we can use the Lipschitz property of the value function to get the result?**
>
> The Lipschitz property states that the value function is bounded by the differences in the underlying dynamics (rewards and transitions). Specifically, for two tasks $M_i$​ and $M_j$​ with their respective value functions $V_{M_i}$​ and $V_{M_j}$​, the difference between the value functions can be bounded by:  $|V^\pi(s_i^+) - V^\pi(s_j^+)| \leq (\text{differences in rewards}) + \gamma (\text{differences in transitions}).$  This means that a small change in rewards or transitions causes a proportionally small change in the value function, governed by the discount factor $\gamma$. The Lipschitz property ensures that differences in rewards and transitions only have a limited impact on the value function due to the contraction property of the Bellman operator. This makes it possible to bound the difference between value functions $|V^\pi(s_i^+) - V^\pi(s_j^+)|$ using the reward and transition metrics. Without the Lipschitz property, there would be no guarantee that the differences between tasks ${M_i}$​ and ${M_j}$ would lead to bounded differences in their value functions.
>
> 4.  **If you use the Lipschitz property of the value function, what is the Lipschitz constant?**
>
> Thank you for your comments. I will provide a brief proof. First, we define the reward model difference, transition model difference, and inverse dynamic difference between tasks as $\epsilon_R$, $\epsilon_T$, and $\epsilon_I$, respectively.
>
> Value function difference:
> $$
> |V^π(s_i^+) - V^π(s_j^+)| = |R(s_i^+, a) - R(s_j^+, a)| + γ | E_{s' ∼ T_i(s_i^+, a)}[V^π(s'^+)] - E_{s' ∼ T_j(s_j^+, a)}[V^π(s'^+)] |
> $$
>
> The expectation term over the transitions can be rewritten:
>
> $$
> | E_{s' ∼ T_i(s_i^+, a)}[V^π(s'^+)] - E_{s' ∼ T_j(s_j^+, a)}[V^π(s'^+)] | ≤ || T_i(s_i^+, a) - T_j(s_j^+, a) ||_1 || V^π ||_∞
> $$
>
> where $\| T_i - T_j \|_1$ measures the difference between the transition distributions and $|| V^π ||_∞$ bounds the value function.
>
> $$
> |V^π(s_i^+) - V^π(s_j^+)| ≤ |R(s_i^+, a) - R(s_j^+, a)| + γ ||T_i(s_i^+, a) - T_j(s_j^+, a)||_1 ||V^π||_∞
> $$
>
> Since the value function is recursive, the Bellman operator propagates the differences at every step. Assuming the reward discrepancy is bounded by \( \epsilon_R \) and the transition dynamics discrepancy by \( \epsilon_T \), we can iteratively bound the value function differences:
>
> $$
> |V^\pi(s_i^+) - V^\pi(s_j^+)| \leq \epsilon_R + \gamma \epsilon_T \| V^\pi \|_\infty.
> $$
>
> The value function difference propagates over multiple steps, scaled by $\gamma$ at each iteration. Using the contraction property of the Bellman operator, the total difference converges geometrically:
>
> $$
> ||V^π(s_i^+) - V^π(s_j^+)||_∞ ≤ \frac{(ε_R + γ ε_T R_m)}{(1 - γ)}
> $$
>
> Include inverse dynamics:
>
> $$
> ||V^π(s_i^+) - V^π(s_j^+)||_∞ ≤ \frac{(ε_R + γ (ε_T+ε_I) R_m)}{(1 - γ)}
> $$
>
> The Lipschitz constant for the value function difference is:
>
> $$
> L = \epsilon_R + \frac{\gamma (ε_T + ε_I) R_m}{1 - \gamma}.
> $$
>
> For Theorem 2, we have provided a more detailed proof in the revised manuscript.

---

> ### Author Response · Authors · 2024-11-25
>
> 5. **Discussion on Baseline Selection**
>
> We appreciate your suggestion regarding the inclusion of relevant works (e.g., [1-3]). We fully understand and value the importance of incorporating more recent methods into the baseline comparisons. However, after our investigation, we found that the implementation code for [1][2] is not publicly available. Additionally, [3] does not belong to the category of context-based meta-RL methods. As a result, it is not feasible to include these methods as baselines for quantitative comparison. Nevertheless, we will add a discussion of these methods in the revised version and provide a detailed analysis of how they differ from our approach. In our paper, we compare against baselines such as VariBAD, PEARL, MetaCURE, and HyperX, all of which are highly influential works within the meta-RL field. Our method shows significant performance improvements over these baselines, especially in the context of out-of-distribution sparse reward task adaptation.
>
> To evaluate the performance of the algorithm in extremely sparse reward scenarios, we chose to use panda-gym. Previous meta-RL algorithms have not used panda-gym. Its environments only provide a reward signal after successfully achieving the task goal. Additionally, the state space and action space in panda-gym are significantly larger, closely resembling real-world robotic arm simulations. This makes panda-gym more suitable for testing the algorithm's exploration efficiency and its adaptability in sparse reward settings.
>
>  6. **Why the proposed method can boost the agent's performance in sparse reward environments?**
>
> We have added an explanation of the SimBelief rapid adaptation mechanism in Section 4, and provided a detailed analysis in Appendix F.  “SimBelief, through the latent task belief,essentially learns the transfer relationships between knowledge across different tasks. In extremely sparse reward scenarios, once the agent succeeds in one task, this prior knowledge of success can be quickly propagated to other similar tasks via the latent task belief, enabling the agent to rapidly adapt to similar types of tasks. A more detailed discussion is provided in Appendix F.”
>
> We hope these revisions address your concerns and clarify the contributions of our work. Thank you again for your constructive feedback, which has been invaluable in improving our submission.

---

> > ### Comment · Reviewer_z5au · 2024-11-26
> >
> > I'd like to thank the authors' detailed response. Below is my reply:
> >
> > - There are still several more new baselines for meta RL, especially for sparse reward, like [1] with code in the link https://github.com/zoharri/mamba. Comparing with there more new and SOTA methods will make this paper more solid. I understand the discussion period is limited, so I hope a detailed comparision in the future revised versions.
> >
> > - As for the sparse reward environment, some work has discussed that zero-shot generalization (a special case of meta RL) is extremly difficult or even impossible when the reward is extremly sparse (Appendix B.7 in [2]). So I'm curious about that, as a  general meta RL method (few-shot adaptation), how the proposed method can improve the performance in sparse reward environments, and how many trajectories in the adaption stage is required? More discussion will make this work more solid.
> >
> > Overall, most of my concerns has been addressed, I have raised my scores into 6.
> >
> >
> > Ref:
> >
> > [1] MAMBA: an Effective World Model Approach for Meta-Reinforcement Learning
> >
> > [2] Task Aware Dreamer for Task Generalization in Reinforcement Learning

---

> ### Author Response · Authors · 2024-11-26
>
> Thank you for your thoughtful follow-up comments and for revisiting our manuscript. We greatly appreciate your feedback and are pleased that most of your concerns have been addressed. Below, we provide additional responses to the points you raised.
>
> **Regarding the first point:**
>
> We appreciate your suggestion to include more recent baselines such as MAMBA. We will include comparisons with such sparse reward-focused baselines in future revised versions to make our results more comprehensive.
>
> **Regarding the second point:**
>
> As highlighted, zero-shot generalization in extremely sparse reward environments is particularly challenging. Algorithms based on the BAMDP framework, such as VariBAD, HyperX, and SimBelief, are capable of achieving strong zero-shot generalization. From Figure 4 and Figure 14 in this paper, it can be observed that SimBelief achieves a high return within the first episode (i.e., a single complete trajectory), which is one of the advantages of the BAMDP framework. In contrast, posterior sampling-based algorithms like PEARL and MetaCURE require interacting with the environment for several episodes before adapting to it.
>
> Our proposed algorithm, SimBelief, learns the latent dynamics of different tasks and distinguishes tasks in the latent space. This approach significantly improves the agent's reasoning ability in unknown environments. In Appendix F.2, we visualize the correlations of specific task belief and latent task belief across different tasks. Specifically, **specific task belief** focuses more on local information between tasks, while **latent task belief** captures the global characteristics of task distributions. The agent can make relatively accurate inferences about the current unknown environment based on these two types of beliefs.
>
> Thank you once again for your detailed comments and for raising your score. We are committed to addressing your suggestions in future iterations and enhancing the work's overall robustness and clarity.

---

### Official Review · Reviewer_cdUU · 2024-10-28

**Soundness:** 2
**Presentation:** 1
**Contribution:** 1
**Rating:** 6
**Confidence:** 4

**Summary:**

The authors propose a novel meta-RL method tackling the problem of efficient task identification in sparse reward settings, where methods that do reward reconstruction are not sufficient.  In a context-based meta-RL framework, they propose inferring the latent task representation through a Gaussian mixture of a variational latent representation and a new task belief similarity latent representation the authors introduce.  This task belief similarity latent is trained to model a  bisimulation-inspired latent task belief metric between two tasks.  They demonstrate modest improvements compared to prior methods in sparse reward, simulated locomotion and manipulation tasks and robustness to OOD task variations.

**Strengths:**

* Introducing ideas of modeling task similarity to meta-RL is interesting and worth exploring
* Sound experimental setup and results

**Weaknesses:**

The biggest weakness is that the proposed method lacks motivation and reasoning about how they achieve the claims the authors make.
* Combining the variational task belief $z_r$ with their proposed task belief similarity $z_l$ through a Gaussian mixture.  This does not combine both types of task representations, but instead samples from one or the other.  Furthermore, $z_r$ already models dynamics information in order to reconstruct the trajectory, so it's unclear what information $z_l$ adds. This is also flawed because the task belief similarity is not modeled as a distribution (at least based on Section 3.2).
* There are a couple things in Section 3.3 that are mentioned briefly but not explained: using the Q-function to train offsets for the task similarity distributions and minimizing KL divergence to the variational task belief $z_r$ when predicting $z_l$.  These details seem important to the method and the fact that the KL divergence is required to “ensure the agent does not confuse similar tasks” indicate that $z_l$ may not be as effective as the authors claim.

Other major weaknesses include:
* Clarity of writing.  The method section, especially 3.2, was difficult to understand.  The experiments section does not describe what the different tasks are in each environment, and Figures 4 and 5 are hard to interpret.
* Insufficient discussion of relevant related work, especially MetaCURE, which seems to tackle the exact problem of task ID with sparse rewards.
* Insufficient discussion of results, especially providing insight into why comparison methods may under or over perform.
* No ablations provided.

**Questions:**

* Why is the inverse dynamics required in Definition 2?  Have you done any ablations for how it compares to the bisimulation metric?
* What are the offsets mentioned in section 3.3 and how are they trained?

---

> ### Author Response · Authors · 2024-11-25
>
> We sincerely thank the reviewer for their detailed feedback and constructive comments. Below, we provide a point-by-point response to the highlighted weaknesses and questions while addressing how we have clarified and improved the manuscript based on these concerns.
>
> **Weaknesses**
>
> 1. Motivation and Reasoning for Achieving Claims
>
> We understand the reviewer's concern regarding the clarity of motivation and reasoning behind our claims. To address this, we have updated the manuscript to:
> * Clearly explain how SimBelief’s latent task belief metric captures the common structure across tasks, which is essential for adaptation in sparse reward environments (Section 3.2).
> * Provide detailed reasoning on how the Gaussian mixture of $b_l$ (latent task belief) and $b_r$ (specific task belief) enables SimBelief to balance global task similarity with specific task dynamics. This combination enhances both exploration and task-specific adaptation, as shown in our experiments (Section 4, Figures 3–5, Appendix F).
>
> 2. Combining $z_r$ with $z_l$ through Gaussian Mixture
>
> In the paper, **we combine $b_r$ and $b_l$, rather than $z_r$ and $z_l$ (Equation 7).** $z_r$ and $z_l$ are sampled from $b_r$ and $b_l$, respectively. We utilize the **overall distribution characteristics** (including both the mean and variance) obtained from the Gaussian mixture of the specific task belief $b_r$ and the latent task belief $b_l$, without the need to sample from the mixed distribution. **$z_l$ represents the sample from latent task belief $b_l$ in the latent space and captures the similarity between different tasks.**(Section 3.1, 3.2, Appendix F) All task distributions share a common latent dynamic space, with different tasks distinguished by different $z_l$, where $z_l$ contains the dynamic similarity information between any two tasks(Equation 8).
>
> 3. KL divergence is required
>
> In the early stages of training, the latent task belief enhances the agent's ability to identify tasks and explore efficiently in the real environment, providing a high-level understanding of the overall task distribution (Appendix F.2). However, for the algorithm to converge stably, it need to incorporate finer-grained information about specific tasks, which is captured in the specific task belief $b_r$. Therefore, we minimize the discrepancy between the latent belief $b_l$ and the specific task belief $b_r$ when training $ \psi_l $.
>
> 4. Using the Q-function to train offsets for the task similarity distributions
>
> Because during the modeling process in the latent space, the latent task belief cannot directly participate in the optimization of the Q-function to ensure the accuracy of the latent dynamics. Instead, during SAC training, exploration occurs in the real environment, and the Q-function acts as a belief shift signal. This signal indirectly influences the latent task belief, causing the latent space to align with the optimization direction of SAC as a whole.
>
> 5. Clarity of writing.
>
> We have revised certain expressions in Section 3.2 of the manuscript. Additionally, we provide detailed descriptions and settings for each task in Appendix E. Figures 4 and 5 are commonly used and effective analytical methods in meta-RL[1][2][3] .
>
> 6. Insufficient discussion of relevant related work, especially MetaCURE.
>
> We provide a detailed discussion of the baselines and related works in Appendix D and Section 5. One drawback of MetaCURE compared to our approach is its reliance on task IDs during the training phase, which can impair the agent's reasoning ability during adaptation. As shown in the experimental results in Figures 3-6, MetaCURE underperforms SimBelief.
>
> 7. Insufficient discussion of results, especially providing insight into why comparison methods may under or over perform.
>
> In Section 3.4, we provide a theoretical analysis of the latent task belief's transferability across different tasks. Furthermore, in Appendix F, we offer a detailed explanation from the perspective of task belief representation, revealing the underlying mechanisms that enable the agent to achieve robust OOD generalization during both the training and adaptation phases.
>
> 8. No ablations provided.
>
> We provide three ablation studies in Appendix G: latent dynamics ablation, inverse dynamics ablation, and ablation on $w_r$ and $w_l$.

---

> > ### Author Response · Authors · 2024-11-25
> >
> > **Questions**
> >
> > 1. Why is the inverse dynamics required in Definition 2? Have you done any ablations for how it compares to the bisimulation metric?
> >
> > In Appendix B, we prove the latent transfer bound of the latent task belief metric, showing that incorporating the inverse dynamics module broadens the range of task transfer distributions. In Appendix G.2, we test the impact of the algorithm with and without the inverse dynamics module on OOD task adaptation. The results demonstrate that the inverse dynamics module enhances the agent's reasoning ability for unknown tasks, further validating the effectiveness of Theorem 2.
> >
> > 2. What are the offsets mentioned in section 3.3 and how are they trained?
> >
> > The offsets in Section 3.3 adjust task similarity distributions to align the latent task belief with the Q-function during SAC training. They are trained indirectly, as the Q-function serves as a belief shift signal, refining the latent task belief through real-environment exploration.
> >
> > [1]Rakelly et al. Efficient off-policymeta-reinforcement learning via probabilistic context variables. ICML, 2019.
> >
> > [2]Zintgraf et al, Varibad: A very good method for bayes-adaptive deep rl via meta-learning. ICLR, 2020.
> >
> > [3]Zintgraf et al, Exploration in approximate hyper-state space for meta reinforcement learning. ICML, 2021.
> >
> >
> > We thank you for your feedback and hope we've addressed your comments adequately. We are happy to answer any further questions and incorporate any further suggestions.

---

> > > ### Comment · Reviewer_cdUU · 2024-11-27
> > >
> > > Thank you for the detailed response and revisions to the paper.  I still have some concerns and questions about the method.
> > >
> > > 1. Combining $b_r$ and $b_l$.
> > > I still have my original concern that using a Gaussian mixture is not combining the information from both types of representations but just sampling from one or the other.
> > >
> > >
> > > 2. What is $b_l$'s objective function?  From the paper, I thought it was Equation 5 so I was confused how you're getting a belief distribution.
> > >
> > >
> > > 3. KL divergence and Q-function offsets.
> > > My main concern is that these components are mentioned very briefly and not fully described at the end of the methods section and seem key to the policy performance.  I am worried that the specific task belief may not actually be learned well and utilized in the way the authors claim if it requires so many adjustments.
> > >
> > >
> > > 4. KL divergence.
> > > The author's reasoning that the policy might benefit more from a high level task representation at the beginning of training is fair, but then using a KL minimization still does not make sense since it moves the specific task belief towards the latent task belief all the time.  Also this objective may be conflicting with $b_l$'s training objective and causing it to model latent task belief instead.
> > >
> > >
> > > 5. What is the specific training objectives for the Q-function offsets?  How critical are these offsets?  How much do they change the beliefs?

---

> ### Author Response · Authors · 2024-11-28
>
> Thank you for your thoughtful and insightful comments. We greatly appreciate your feedback and have made several updates in the revised manuscript to address your concerns.  I will provide a detailed answer to your question from the perspective of technical implementation details:
>
> **1. Combining $b_r$​ and $b_l$:**
>
> At the code level, we model $b_r$ and $b_l$ as mean and logvar (i.e., the output of $\psi_r$ represents the mean and variance of $b_r$, and the output of $\psi_l$ represents the mean and variance of $b_l$), which are used to represent these two Gaussian distributions, similar to the belief modeling approach in **VariBAD** [1]. We sample $z_r$ from $b_r$ to reconstruct $p_\phi$. On the other hand, we sample $z_l$ from $b_l$, and after optimization with Equation 8, $b_l$ can capture latent task belief similarity information. Until now, we have been sampling from both Gaussian distributions (i.e., $b_r$ and $b_l$) to represent different types of information. **We then mix the two distributions in Equation 7, which, at the code level, is also represented as the mean and logvar of $b$. We do not need to perform any sampling operation on $b$, instead, we treat $b$ (including its mean and variance) as a whole**, combine it with $s$ to form an augmented state, and feed the augmented state as input to the policy.
>
> **2. What is $b_l$'s objective function?**
>
> As shown in Figure 2, $b_l$ is the output of $\psi_l$, and we optimize $\psi_l$ using Equation 8. From the code perspective, the input to $\psi_l$ is $h$ (historical information), and the output is the mean and variance of $b_l$.
>
> **3. KL divergence.**
>
> Regarding the use of KL divergence, I would like to clarify in detail that we use KL divergence to regularize the latent task belief $b_l$ toward the specific task belief $b_r$, not the other way around. The KL loss is used to regularize $b_l$, making it incorporate information from the specific task, which allows the agent to explore the latent space more purposefully toward the specific task (**Appendix F.1, Figure 8**). This approach is similar to the KL loss regularization used in **DreamerV2** [2].
>
> As for why KL loss continues to participate in the optimization of $b_l$ during the training phase, this depends on the training process of our algorithm. As shown in the pseudocode **(Appendix C)**, $b_l$ is optimized alongside SAC, while $b_r$ is optimized with VAE. For example, in one iteration, SAC is updated 2000 times, while VAE is updated 20 times, the entire training process consists of 1000 iterations. Since the optimizations of $b_l$ and $b_r$ are not synchronized, the KL loss needs to continuously participate in the optimization of $b_l$ during the training phase.
>
> Additionally, the KL loss does not interfere with $b_l$'s ability to capture the global information of the task distribution (**Appendix F.2, Figure 9**).
>
> **4. Regarding the offset**
>
> We directly use the $q$-loss from the SAC algorithm to optimize the offset. To address your concerns, we have added an ablation study on the offset in **Appendix G.4**.
>
> During the training phase, we apply the offset ($\Delta \mu, \Delta \sigma$) to the latent task belief **primarily to improve the stability of the algorithm's convergence in continuous control tasks** (e.g., Cheetah-Vel-Sparse). However, **this does not have a significant impact on the overall performance of the algorithm **(Figure 13)** and does not affect the learned latent task belief $b_l$ (Figure 9)**. Even without using the offset during training, SimBelief still outperforms other baselines.
>
> [1]Zintgraf et al, Varibad: A very good method for bayes-adaptive deep rl via meta-learning. ICLR, 2020.
>
> [2]Hafner et al,Mastering Atari with Discrete World Models. ICLR,2021.
>
> We hope these revisions address your concerns and provide more clarity on the aspects you raised. Thank you once again for your valuable feedback, and we look forward to your continued suggestions.

---

> ### Author Response · Authors · 2024-12-01
> **Thank you for your valuable time**
>
> We greatly appreciate the time and effort you've taken to provide feedback and would be grateful for any additional comments or suggestions you may have. Your insights are incredibly valuable to us as we aim to improve the quality of our work.  We kindly request to reconsider your score if your concerns are sufficiently addressed.
>
>
> If there is anything else you need from us or if you require further clarification, please do not hesitate to let us know. We would love to incorporate any further specific suggestions or concerns you have.

---

> > ### Comment · Reviewer_cdUU · 2024-12-02
> >
> > Thank you for the clarifications and additional experiments, especially with the KL divergence and offset components of the method.  I maintain my concerns about the mixture of the two latent representations and the overall complexity of the method, which were partially addressed with the authors' responses.  I have adjusted my score accordingly.

---

> > > ### Author Response · Authors · 2024-12-03
> > >
> > > We are glad we were able to alleviate your concerns. Thank you for replying promptly and for raising your score. Your thorough review and insightful comments have been instrumental in enhancing the quality of our paper.

---

### Official Review · Reviewer_RXre · 2024-11-04

**Soundness:** 3
**Presentation:** 2
**Contribution:** 3
**Rating:** 6
**Confidence:** 3

**Summary:**

This paper presents a novel framework called SimBelief, which effectively extracts common features of similar tasks using the Bisimulation metric for meta-RL tasks.

**Strengths:**

This paper is generally well-constructed and well-motivated and provides some theoretical background on the proposed method.
In addition, the paper has the potential to help the community understand the use of latent embedding for task generalization.

**Weaknesses:**

The paper includes many symbolic notations, which can confuse readers without strict and coherent representation. For example, in Fig 2, it seems $q_{\phi}$ outputs h, but the notation of Eq. (6) or others in the manuscript, $q_{\phi}$ outputs $z_r$.

Similarly, some of the notations are used before proper definition, which hinders the readers from fully understanding the contents.

The paper's reproducibility is questionable as it consists of various complex components for the framework.

**Questions:**

(1) Regarding readability: In Sec. 2.1., N is not properly defined before use. Inverse dynamics in Definition 2 or Eq.(3) is used before it is properly defined.

(2) Some of the manuscript's functions are unclear. What is the output of inverse dynamics $I_{i}^{\pi}(s_i^{+},s_i^{',+})$ ?
What is the input and output for $q_{\phi}$, $\psi_{l}$, $\psi_{r}$ ?

(3) Similarly, In Algorithm 1, it is unclear $b_{r}^i={q_{\phi}(z_r^i|\tau_{:t})}_{0:T-1}$ means.

(4) What are the $L_{rec}$ and $L_{bisim}$ in Figure 2? If they are defined in the manuscript differently, please use the same notation.

(5) How to determine a proper $w_r$? and how sensitive is it to the overall framework?

(6) The proposed method is complex and contains various components, so presenting only a simple algorithm raises questions about reproducibility. Do the authors plan to release the code for the benefit of the RL community?

---

> ### Author Response · Authors · 2024-11-24
>
> We thank the reviewer for their detailed feedback and insightful comments. Below, we provide a point-by-point response to the weaknesses, questions, and concerns raised in the review.
>
> **Weaknesses**
>
>  **Symbolic Notations:** We acknowledge the concern about the complexity and lack of clarity in some symbolic notations. To address this, we have made the following improvements in the revised manuscript:
>
>    - Added clear definitions of $q_{\phi}$,$\psi_l$,$\psi_r$ and in Section 3.2 before their use, ensuring coherence and clarity in their roles and outputs.
>    - We consider $ q_\phi $ as the forward reasoning process to infer the task belief $ z_r $, corresponding to$z_r \sim \psi_r(b_r \mid h) q_\phi(h \mid \tau_{:t}) $ in the algorithm.
>
> **Questions**
>
> 1. **Regarding $N$ in Sec. 2.1:** In the revised manuscript, we explicitly define $N$ as the number of task episodes within a meta-episode. Additionally, we include a clarifying example to illustrate how $N$ fits into the problem formulation.
> 2. **Role and Output of Inverse Dynamics :** The inverse dynamics predicts the action $a$ required to transition from the current augmented state to the next augmented state. This component is crucial for learning task dynamics and reasoning in the latent space (Appendix G.2.).  Specific task belief: $z_r \sim \psi_r(b_r \mid h) q_\phi(h \mid \tau_{:t}) $;  latent task belief $z_l \sim \psi_l(b_l \mid h) q_\phi(h \mid \tau_{:t}) $.
> 3. **Clarifying :** In Algorithm 1, $b_r^i=q_\phi\left(z_r^i \mid \tau_{1: T}\right)$ refers to the specific task belief inferred by the context encoder using trajectory . This trajectory comprises observed state-action-reward tuples collected during the task. We expanded the explanation in Section 3.3 to ensure clarity. In the latest version, we have modified it to $z_r \sim \psi_r(b_r \mid h) q_\phi(h \mid \tau_{:t}) $.
> 4. **Notations $L_{rec}$ and $L_{bisim}$ in Figure 2:** $L_{rec}$ denotes the reconstruction loss, which ensures accurate latent dynamics reconstruction, while $L_{bisim}$ represents the bisimulation loss, which captures task similarities in the latent space. These terms are now consistently defined in Section 3.2, with detailed descriptions in Appendix F.1.
> 5. **Weight Sensitivity ($w_r$,$w_l$):** The weights determine the balance between specific and latent task beliefs in the Gaussian mixture model. We conducted a sensitivity analysis (Appendix G.3) and observed that $w_r$,$w_l$=0.5,0.5 achieves the best trade-off between exploration efficiency and OOD task adaptation. These results are detailed in the revised manuscript.
> 6. **Reproducibility:** We recognize the complexity of the proposed method and the need for clear reproducibility. To this end, we have included pseudocode for the SimBelief algorithm in Appendix C and provided details on the environment settings in Appendix E. Furthermore, we plan to release the source code upon acceptance to facilitate replication by the community.
>
> We believe these revisions address the concerns raised and further strengthen the clarity and impact of our work. Thank you again for your constructive feedback, which has been instrumental in improving our submission.

---

> > ### Comment · Reviewer_RXre · 2024-11-25
> >
> > Thank you to the authors for their effort in addressing the concerns and clarifying the questions. The authors' response and revised manuscript addressed most of my concerns, and I have adjusted my score accordingly.

---

> > > ### Author Response · Authors · 2024-11-25
> > >
> > > Thank you for your quick reply, and for raising the review score. Please let us know if you have any further questions.

---

### Official Review · Reviewer_6WZh · 2024-11-04

**Soundness:** 4
**Presentation:** 3
**Contribution:** 3
**Rating:** 8
**Confidence:** 5

**Summary:**

The paper uses the bisimulation metrics to measure task similarity and learn the common structure across tasks for rapid task identification and adaptation in meta-RL.

**Strengths:**

- The proposed method is well formulated and clearly presented.
- The effectiveness of the latent task belief metric is validated by theoretical guarantee.
- Experiments demonstrate the superiority of the proposed method over strong baselines, especially the generalization capabilities to OOD testing tasks.

**Weaknesses:**

- The topic of online meta-RL is kind of old.
- The core of the proposed method is using the reward and state transition functions $p(s’,r|s,a)$, or called world model, to measure task similarity in a latent space and hence infer task belief for context-based meta-RL. This kind of task inference has been investigated by existing works like VariBAD.
- The baselines are kind of old, mainly 2019-2021.

**Questions:**

- No major flaws with the paper. The key difference and advantages over VariBAD could be further enhanced and elaborated, since both infer task beliefs using the world model.
- Some notations are not well explained and kind of confusing, e.g., $z_l$ and $z_r$ are confusing, and why use two distributions?
- The proposed method is motivated by the bisimulation metric, which is originally proposed for deterministic MDPs (Castro, 2020). Can the proposed method scale to stochastic MDPs?

---

> ### Author Response · Authors · 2024-11-24
>
> Thank you for the detailed review and valuable feedback. We appreciate the opportunity to clarify and address your comments. Below are our responses to your identified weaknesses, questions, and suggestions.
>
> **Weaknesses**:
>
> 1. "The topic of online meta-RL is kind of old."
>
> While online meta-RL has been extensively studied, our contribution lies in leveraging the latent task belief metric inspired by bisimulation to address the limitations of existing methods in sparse reward settings. Unlike previous approaches, SimBelief utilizes the latent task belief metric to learn the common structure between tasks in the latent space, enabling the agent to exhibit robust and rapid adaptation even in environments with extremely sparse rewards. We propose a novel latent space learning framework. As shown in our experiments (Section 4, Figures 3–5), SimBelief outperforms state-of-the-art methods, particularly in challenging sparse-reward environments, demonstrating its relevance and practical impact. How to improve information utilization efficiency and enhance the agent's online adaptation ability with limited online data remains a meaningful and significant research topic.
>
> 2. "The core of the proposed method is using reward and state transition functions, or a world model, to measure task similarity, which overlaps with existing works like VariBAD."
>
> SimBelief differentiates itself from VariBAD in three critical ways:
>    - **Latent Task Belief Metric**: Instead of relying solely on posterior sampling, our metric integrates inverse dynamics to capture task similarities, which significantly enhances the reasoning and exploration capabilities of the agent.
>    - **Theoretical Contributions**: Our theoretical analysis (Theorems 1 and 2, Appendix B) establishes the conditions under which task similarities translate to policy transferability, providing guarantees for its effectiveness in online meta-RL.
>    - **Experimental Demonstration**: The superior performance on OOD tasks (Figures 3–6) and the visualization of task beliefs (Figure 6) highlight the unique advantages of our approach in capturing global task structures.
>
> 3. "The baselines are kind of old, mainly 2019–2021."
>
>    We acknowledge the age of some baselines but emphasize their continued relevance in the field. For instance, VariBAD, HyperX, and MetaCURE remain widely used benchmarks in meta-RL and exhibit strong and robust performance. Additionally, our method introduces significant advancements over these baselines, as detailed in the experimental results (Section 4 and Appendix F). To further strengthen our comparison, we plan to include newer benchmarks in future work.
>
> **Questions**:
>
>    1."Key differences and advantages over VariBAD could be further elaborated."
>
> SimBelief’s key advantage lies in its latent task belief metric, which captures global task structures more effectively. This is particularly evident in sparse-reward environments, where VariBAD struggles to generalize (Section 4, Figures 3–5). Additionally, SimBelief only reconstructs past trajectories and combines the specific task belief with the latent task belief, enhancing the agent's reasoning ability in unknown environments, making it more adaptable to real-world scenarios.
>
>    2. "Some notations are confusing, e.g., and , and why use two distributions?"
>
> Thank you for highlighting this. represents the latent task belief that captures inter-task similarities, while denotes the specific task belief focused on task-specific details. Their integration balances global structure understanding and local adaptation, as detailed in Sections 3.2–3.3. We have revised the manuscript to clarify this distinction (Appendix F.2).
>
>    3. "Can the proposed method scale to stochastic MDPs?"
>
> Yes, SimBelief is designed to handle stochastic MDPs by leveraging the BAMDP framework. Our latent task belief metric accounts for the stochastic nature of transitions, as demonstrated in both theoretical analysis (Appendix B) and experiments on stochastic environments like Walker-Rand-Params (Section 4).
>
> We thank you for your feedback and hope we've addressed your comments adequately. We are happy to answer any further questions and incorporate any further suggestions.

---

> > ### Comment · Reviewer_6WZh · 2024-11-25
> >
> > Thanks for the detailed response, which has addressed most of my concerns. I still have two questions as follows.
> >
> > - Regarding the difference from VariBAD, "Latent Task Belief Metric: Instead of relying solely on posterior sampling, our metric integrates inverse dynamics to capture task similarities, which significantly enhances the reasoning and exploration capabilities of the agent." The metric contains three parts: the reward function, the forward dynamics, and the inverse dynamics. The proposed method uses the inverse dynamics additionally. Will including the inverse dynamics "significantly" improve the task inference capabilities? Or any empirical evidence with ablation study? Intuitively, the three parts seem to be equally significant.
> >
> > - The paper claims the "significant" performance gain in sparse-reward scenarios and OOD test tasks. As mentioned in the response "Latent Task Belief Metric", the difference from existing work is the inclusion of the inverse dynamics modeling. Does the significant performance gain come from the inverse dynamics modeling, since the existing work has already includes the reward function and forward dynamics modeling?

---

> > > ### Author Response · Authors · 2024-11-25
> > >
> > > We thank the reviewer again for the time and effort of evaluating our paper.
> > >
> > > **Regarding the first question**:
> > >
> > > First, I would like to clarify the main difference between our approach and VariBAD. VariBAD focuses solely on reconstructing specific tasks without considering the relationships between tasks. In contrast, our method models dynamics in the latent space and leverages latent dynamics to learn the correlations between different tasks.
> > >
> > > Second, the reason we include the inverse dynamics in our defined latent task belief metric is to address scenarios with extremely sparse reward signals. For example, in various tasks within panda-gym, a reward is only received upon the successful execution of the task. In such cases, it is crucial to model the task dynamics more accurately in the latent space to learn more effective task structures, thereby enabling the latent task belief to capture task similarity information more precisely. This motivation is the primary driver behind our design of the latent task belief metric. Additionally, we have conducted ablation experiments to verify the effectiveness of inverse dynamics for OOD task generalization. As shown in Appendix G.2, removing the inverse dynamics from the latent task belief metric leads to a decrease in adaptation performance.
> > >
> > > **Regarding the second question**:
> > >
> > > The superiority of SimBelief compared to current methods lies in its ability to learn task similarity information in the latent dynamic space. By incorporating the inverse dynamics in the latent task belief metric, as demonstrated in the OOD task adaptation experiments in Appendix G.2, SimBelief achieves significantly better results. This aligns with Theorem 2, which shows that inverse dynamics can broaden the transfer range between tasks, validating our expectations. The inverse dynamics play a complementary role in enhancing the learning of latent dynamics.
> > >
> > > We hope these revisions address your concerns and clarify the contributions of our work.

---

> > > > ### Comment · Reviewer_6WZh · 2024-11-26
> > > > **Thanks for your detailed response.**
> > > >
> > > > I realized that I might have misunderstood the proposed method a bit. Since both the proposed method and VariBAD used the world model (i.e., the reward and state transition functions) to infer task belief, I thought they were very similar. After I read the authors' response and revised manuscript carefully, I realized that there is a significant difference between them.  From what I understand, VariBAD models task similarity **implicitly** in the latent space, while the proposed method does that **explicitly**. Also, the techniques for deriving task beliefs are also different.
> > > >
> > > > So, the paper reveals good technique novelty, sound theoretical guarantee, and comprehensive experimental results. I will raise my score to 8. I am looking forward to future work on more recent topics, such as offline meta-RL, in-context RL.

---

> > > > > ### Author Response · Authors · 2024-11-26
> > > > >
> > > > > Thank you for your thoughtful feedback and for revisiting our manuscript with such care. We truly appreciate your acknowledgment of the distinction between our method and VariBAD, especially regarding the explicit modeling of task similarities in the latent space using the latent task belief metric and the different techniques for deriving task beliefs.
> > > > >
> > > > > We are glad that our response and the revised manuscript clarified these differences and highlighted the novelty and theoretical contributions of our work. Your recognition of these aspects and your encouragement mean a lot to us.
> > > > >
> > > > > We are also grateful for your constructive comments on exploring recent directions such as offline meta-RL and in-context RL. These are indeed exciting areas, and we plan to incorporate these perspectives into our future research.
> > > > >
> > > > > Thank you once again for raising your score and for your support of our work. We hope to continue building on these ideas in future endeavors.

---

### Meta-Review · Area_Chair_BB5p · 2024-12-15

**Metareview:**

The authors propose a meta-reinforcement learning framework based on bisimulation metrics to compute similarity of task belief from a Bayesian perspective. They infer the latent task representation through a Gaussian mixture of a variational latent representation and a new task belief similarity latent representation that models a bisimulation-based task belief metric. They demonstrate some improvements over prior methods in sparse reward, simulated locomotion and manipulation tasks and show robustness to OOD task variations.

Reviewers found the contribution of using task similarity in meta-RL novel and interesting, and appreciated the theoretical background. However, the clarity of the writing can be improved, especially the notation. Because the framework is fairly complex, there is concern about reproducibility of the method.

Overall, there were some weaknesses to the paper that were satisfactorily addressed in the rebuttal phase. The use of task belief similarity metrics for meta-RL and the corresponding theoretical analysis and good empirical results are a significant enough contribution to warrant acceptance.

**Additional Comments On Reviewer Discussion:**

In the initial reviews, there were concerns about mistakes in the theoretical contributions raised by reviewer z5au. However, in the rebuttal phase they were satisfactorily addressed. Reviewer cdUU also had questions about various design choices in SimBelief, which authors designed additional experiments and presented results to address. In the end, reviewers unanimously agreed to accept this paper.

---

### Decision · Program_Chairs · 2025-01-22

Accept (Poster)